# Transient oxytocin signaling primes the development and function of excitatory hippocampal neurons

Silvia Ripamonti[1,2], Mateusz C Ambrozkiewicz[1,3], Francesca Guzzi[2,4], Marta Gravati[5], Gerardo Biella[5], Ingo Bormuth[3], Matthieu Hammer[1], Liam P Tuffy[1], Albrecht Sigler[1†], Hiroshi Kawabe[1], Katsuhiko Nishimori[6], Mauro Toselli[5], Nils Brose[1], Marco Parenti[2,4*], JeongSeop Rhee[1*]

[1]Department of Molecular Neurobiology, Max Planck Institute of Experimental Medicine, Göttingen, Germany; [2]Department of Medicine and Surgery, University of Milan-Bicocca, Monza, Italy; [3]Cortical Development, Institute of Cell Biology and Neurobiology, Charité-Universitätsmedizin, Berlin, Germany; [4]NeuroMi - Milan Center for Neuroscience, Monza, Italy; [5]Department of Biology and Biotechnology, University of Pavia, Pavia, Italy; [6]Department of Molecular and Cell Biology, Graduate School of Agricultural Science, Tohoku University, Miyagi, Japan

**\*For correspondence:** marco. parenti@unimib.it (MP); rhee@em. mpg.de (JSR)

**Present address:** [†]Science Products GmbH, Hofheim am Taunus, Germany

**Competing interests:** The authors declare that no competing interests exist.

**Abstract** Beyond its role in parturition and lactation, oxytocin influences higher brain processes that control social behavior of mammals, and perturbed oxytocin signaling has been linked to the pathogenesis of several psychiatric disorders. However, it is still largely unknown how oxytocin exactly regulates neuronal function. We show that early, transient oxytocin exposure *in vitro* inhibits the development of hippocampal glutamatergic neurons, leading to reduced dendrite complexity, synapse density, and excitatory transmission, while sparing GABAergic neurons. Conversely, genetic elimination of oxytocin receptors increases the expression of protein components of excitatory synapses and excitatory synaptic transmission *in vitro*. *In vivo*, oxytocin-receptor-deficient hippocampal pyramidal neurons develop more complex dendrites, which leads to increased spine number and reduced γ-oscillations. These results indicate that oxytocin controls the development of hippocampal excitatory neurons and contributes to the maintenance of a physiological excitation/inhibition balance, whose disruption can cause neurobehavioral disturbances.

## Introduction

Oxytocin (Oxt) is produced by magnocellular neurons of the paraventricular (PVN) and supraoptic nuclei (SON) of the hypothalamus and released into the bloodstream from posterior pituitary nerve terminals. By activating G protein-coupled Oxt receptors (Oxtr) in peripheral target tissues, Oxt exerts multiple effects, most prominently uterine contraction during parturition and milk ejection after delivery. In addition, Oxt is released in the brain by synaptic release from projections of parvo-cellular PVN neurons and by hormone-like release from the somata and dendrites of magnocellular neurons (*Ludwig and Leng, 2006*). Based on these processes and its long half-life, Oxt can act in diverse Oxtr-expressing brain areas even when direct innervation by Oxt-containing fibers is absent (*Stoop, 2012*).

Key regulatory effects of Oxt on animal and human behavior include promotion of facial recognition and maternal nurturing (*Ferguson et al., 2001*, *2000*), control of anxiety (*Bale et al., 2001*; *Yoshida et al., 2009*), and regulation of spatial memory (*Tomizawa et al., 2003*). These functions

are corroborated by studies on mice lacking Oxt or Oxtrs (*Nishimori et al., 1996*). Notably, the behavioral phenotype of *Oxtr* knock-out ($Oxtr^{-/-}$) mice recapitulates several symptoms of autism spectrum disorders (ASDs) (*Lee et al., 2008*; *Sala et al., 2011*). Conversely, behavioral changes in mouse ASD models that are not based on direct perturbations of Oxt signaling, such as µ opiod receptor (*Opmr1*) (*Gigliucci et al., 2014*) or contactin associated protein-like 2 (*Cntnap2*) knock-outs (*Peñagarikano et al., 2015*), are rescued upon Oxt administration, and even human ASD patients improve with Oxt treatment (*Andari et al., 2010*; *Hollander et al., 2007*; *Striepens et al., 2011*). Moreover, *Oxtr* polymorphisms (*LoParo and Waldman, 2015*) and abnormal methylation of the *Oxtr* promoter (*Elagoz Yuksel et al., 2016*) have been linked with ASDs, supporting the notion that altered Oxt signaling can contribute to ASDs.

At delivery, an Oxt surge in the maternal bloodstream promotes labor and subsequent lactation. The circulating Oxt reaches the fetal brain, where it regulates the developmental switch of GABAergic signaling from excitation to inhibition (*Tyzio et al., 2006*). This switch appears to be impaired in the valproate-intoxication and fragile-X-syndrome mouse models of ASDs, and to be restored by Oxt treatment. Moreover, administration of an Oxtr antagonist to naïve pregnant mice one day before delivery leads to ASD-like features in the offspring (*Tyzio et al., 2014*). These findings indicate that an early blockade of Oxt signaling affects the development of synaptic wiring in the brain and leads to lifelong consequences, including neuropsychiatric diseases. This notion is supported by the observations that $Oxtr^{-/-}$ mice show an increased seizure propensity and that the relative contribution of GABAergic synapses is reduced in cultured hippocampal $Oxtr^{-/-}$ neurons (*Sala et al., 2011*). However, the hypothesis that early, transient Oxt signaling shapes or 'primes' neuronal morphology and function in the brain remains to be tested.

We attempted in our study to mimic the maternal Oxt surge prior to and around parturition and its effect on brain development in the offspring by studying early effects of transient Oxt signaling on developing neurons *in vitro*, and by analyzing related consequences of Oxtr loss *in vivo*. We demonstrate that Oxt exerts an early and cell-type specific 'priming' effect on developing excitatory neurons to regulate dendrite branching and synapse development, synaptic transmission, and the synchronicity of neuronal networks. These findings are likely of key relevance for the molecular pathology of brain diseases that involve altered Oxt signaling.

## Results

### Oxt reduces dendrite branching and function of glutamatergic neurons via $G_{q/11}$-coupled Oxtrs

In mice, a surge of Oxt appears during the last days of gestation (*Pfaff et al., 2002*) and peaks just before delivery (*Gimpl and Fahrenholz, 2001*). To assess the effects of transient exposure of hippocampal neurons to Oxt, we reenacted *in vitro* the *in vivo* Oxt maternal surge by treating autaptic cultures (*Bekkers and Stevens, 1991*; *Burgalossi et al., 2012*) with 100 nM Oxt for 1–3 days *in vitro* (DIV), starting on the first day after seeding. Oxt exposure did not affect neuronal survival (*Figure 1—figure supplement 1a,b*) but reduced the dendritic arborization of neurons at 14 DIV, as assessed by Sholl analysis (*Sholl, 1953*) of cells stained with an antibody to the dendrite marker Map2 (*Figure 1a–c*). The altered dendrite branching was found only in wild-type (WT) glutamatergic neurons, but not in WT hippocampal inhibitory neurons identified by VGAT staining (*Figure 1—figure supplement 2a,b*) or in *Oxtr*-deficient neurons (*Figure 1—figure supplement 3a–c*). To test whether the altered morphology affects synaptogenesis, we quantified the number of excitatory synapses of hippocampal autaptic neurons. Using antibodies to VGLUT1 and PSD95 to label glutamatergic pre- and post-synaptic compartments, respectively (*Nair et al., 2013*), we determined the number of immunoreactive puncta in each Map2-labeled neuron (*Figure 1d*). Oxt exposure reduced the number of juxtaposed glutamatergic pre- and post-synaptic sites, but no changes were detected in the spatial coincidence of pre- and postsynaptic labeling (Mander's overlapping coefficient) (*Figure 1e–g*), indicating that the integrity of the remaining glutamatergic synapses is maintained.

Oxt exerts its physiological effects via the Oxtr, which is expressed in cultured hippocampal neurons already at DIV1 (*Leonzino et al., 2016*), but due to the high homology of Oxt and arginine vasopressin (AVP), Oxt can also act as a partial agonist on AVP receptors (Avprs) (*Gimpl and Fahrenholz, 2001*). To test if Oxtrs mediate the Oxt effects seen in our system, we exposed autaptic

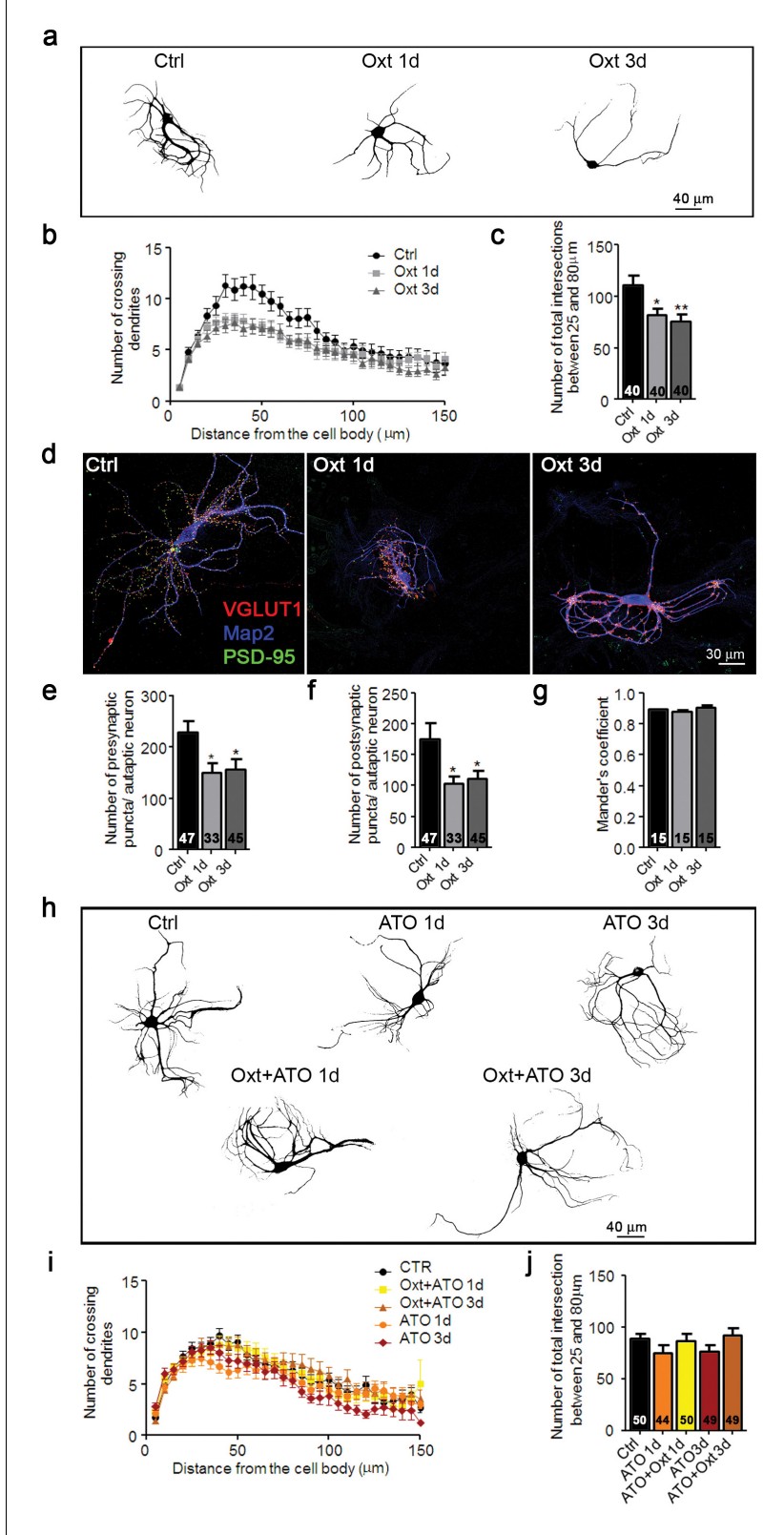

**Figure 1.** Oxt-treatment impairs dendrite development and reduces synapse numbers in glutamatergic autaptic hippocampal neurons. (a) Representative binary images of neurons that had been exposed to 100 nM Oxt for 1 (Oxt 1d) or 3 days (Oxt 3d) after plating and immunostained at 14DIV using an anti-Map2 antibody. (b) Sholl analysis of Ctrl neurons and neurons treated with Oxt 1d or Oxt 3d. Numbers of cells for quantification are shown

*Figure 1 continued on next page*

*Figure 1 continued*

in panel (c). (c) Average numbers of dendrites intersecting Sholl circles at 25–80 µm distance from the cell body in Ctrl neurons and neurons treated with Oxt 1d or Oxt 3d (*p<0.05, Ctrl vs, Oxt 1d; **p<0.01, Ctrl vs. Oxt 3d). (d) Representative images of Ctrl neurons and neurons treated with Oxt 1d or Oxt 3d, fluorescently stained with antibodies directed against VGLUT1, PSD95, and Map2 at 14DIV. (e and f) Average numbers of VGLUT1-positive glutamatergic presynapses (E), and PSD95-positive postsynapses (F) in Ctrl neurons and neurons treated with Oxt 1d or Oxt 3d (*p<0.05, Ctrl vs. Oxt 1d and Oxt3d). (g) Mander's coefficient measures of overlap between VGLUT1 and PSD95 labeling in Ctrl neurons and neurons treated with Oxt 1d or Oxt 3d. (h) Representative binary images of neurons immunofluorescently labeled for Map2 from control cultures (Ctrl) and cultures treated with 100 nM atosiban alone (ATO), or with 100 nM atosiban plus 100 nM Oxt (ATO + Oxt) for 1 day (1d) or 3 days (3d). (i) Sholl analysis of Ctrl neurons and neurons treated with 100 nM ATO, or ATO + Oxt for 1d or 3d. Number of cells are shown in panel (J). (j) Average numbers of dendrites intersecting Sholl circles at 25–80 µm distance from the cell body in Ctrl neurons and neurons treated with 100 nM ATO, or with ATO + Oxt for 1d or 3d. Ctrl, control. Data are shown as mean ± SEM. Numbers of analyzed cells are indicated in the histogram bars. Statistical analyses were performed with one-way ANOVA followed by post-hoc Bonferroni test. See Table in *Supplementary file 2*.

The following figure supplements are available for figure 1:

**Figure supplement 1.** Oxt-treatment does not affect survival of primary hippocampal neurons.

**Figure supplement 2.** Oxt-treatment does not alter the dendrite development of GABAergic autaptic hippocampal neurons.

**Figure supplement 3.** Oxt-treatment does not alter the dendrite development and function of Oxtr- deficient autaptic hippocampal neurons.

cultures to Oxt for 1 or 3 DIV in the presence of the Oxtr antagonist Atosiban (ATO). Morphological analyses at 14 DIV showed that ATO treatment abolished the Oxt-induced effects on dendrite arborization, whereas ATO alone was ineffective (*Figure 1h–j*), indicating that the Oxt effects on dendrites and synapses of hippocampal neurons are mediated by Oxtrs.

To test whether the morphological changes caused by Oxt contribute to alterations in synaptic transmission, we performed patch-clamp recordings at 9–14 DIV in Oxt-treated hippocampal autaptic neurons. The amplitudes of evoked excitatory postsynaptic currents (EPSCs) were decreased in Oxt-treated glutamatergic neurons (*Figure 2a,c*). The same was true for the size of the apparent readily releasable pool (RRP) of glutamatergic synaptic vesicles (SVs), as assessed by the EPSC induced by the application of a hypertonic sucrose solution (*Nair et al., 2013*; *Rosenmund and Stevens, 1996*) (Figure a,d). In contrast, neither evoked inhibitory postsynaptic currents (IPSCs) amplitudes nor the RRP of GABAergic neurons, which represent a minor cell fraction in hippocampal autaptic cultures, were affected by Oxt, indicating that Oxt exclusively acts on excitatory hippocampal neurons in culture (*Figure 2b–d*). The vesicular release probability ($P_{vr}$) of glutamatergic and GABAergic neurons, calculated by dividing the postsynaptic charge transfer elicited by an action potential (AP) by the postsynaptic charge transfer in response to RRP release, was not changed by Oxt exposure (*Figure 2e*), indicating that the fusion-probability of primed SVs is not altered by Oxt. We next studied synaptic short-term plasticity (STP) by measuring synaptic responses during 10 Hz stimulus trains, and found that EPSC and IPSC amplitudes depressed similarly and irrespective of Oxt-treatment (*Figure 2l,m*). These findings indicate that Oxt specifically causes strong reductions of EPSCs but does not affect the fundamental characteristics of the presynaptic transmitter release machinery.

Miniature EPSC (mEPSC) amplitudes were not affected by Oxt treatment, indicating that postsynaptic sensitivity to quantal release at single synapses is not affected by Oxt (*Figure 2i,j*). In contrast, we detected a significant reduction of mEPSC frequency, indicating that Oxt reduces the number of functional glutamatergic synapses (*Figure 2k*). Again, reflecting the morphological and IPSC data, mIPSC events in GABAergic neurons were not altered by Oxt (*Figure 2i–k*).

To further assess the selectivity of Oxt effects on excitatory neurons, we quantified ionic currents upon exogenous application of glutamate or GABA. Corroborating our PSC data, GABA-induced currents were not changed in Oxt-treated GABAergic neurons, whereas Oxt-exposed glutamatergic

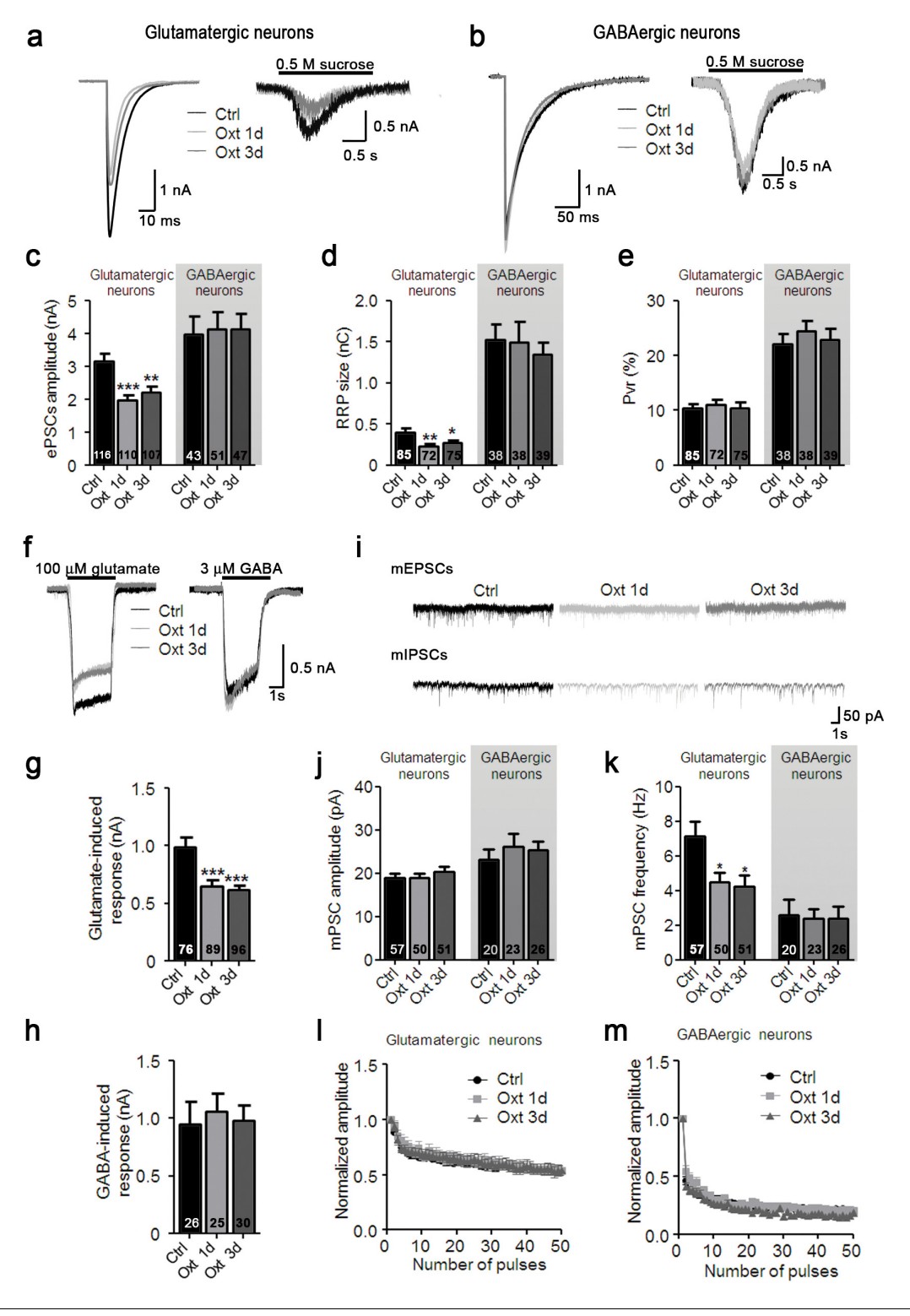

**Figure 2.** Oxt-treatment reduces evoked and spontaneous glutamatergic synaptic transmission in autaptic hippocampal neurons. (a) Representative traces of evoked EPSCs (left) and postsynaptic currents evoked by 0.5 M sucrose (right) from control glutamatergic autaptic neurons (Ctrl, black) and from glutamatergic autaptic neurons treated with Oxt for 1 day (Oxt 1d, light grey) or 3 days (Oxt 3d, dark grey). (b) Representative traces of evoked IPSCs (left) and postsynaptic currents evoked by 0.5 M sucrose (right) from control GABAergic autaptic neurons (Ctrl, black) and from GABAergic autaptic neurons treated with Oxt 1d (light grey) or 3d (dark grey). (c–e) Average

*Figure 2 continued on next page*

*Figure 2 continued*

values of evoked PSC amplitudes (C; **p<0.001, Ctrl vs. Oxt 3d; ***p<0.0001, Ctrl vs. Oxt 1d), (d) apparent readily releasable vesicle pool (RRP) size (**p<0.001, Ctrl vs. Oxt 1d; *p<0.05, Ctrl vs. Oxt 3d), and eE) $P_{vr}$ in Ctrl neurons and neurons treated with Oxt 1d or Oxt 3d. The RRP was measured as the net charge transferred during application of 0.5 M sucrose solution. (f) Representative traces of responses of glutamatergic neurons to exogenous glutamate (100 µM, left) and of GABAergic neurons to exogenous GABA (3 µM, right). Data are from Ctrl and neurons treated with Oxt 1d or Oxt 3d. (g) Average amplitudes of EPSCs induced by 100 µM glutamate in glutamatergic Ctrl neurons and neurons treated with Oxt 1d or Oxt 3d (***p<0.0001, Ctrl vs. Oxt 1d and Oxt 3d). (h) Average amplitudes of IPSCs induced by 3 µM GABA in GABAergic Ctrl neurons and neurons treated with Oxt 1d or Oxt 3d. (i) Representative traces of mEPSCs (top), and mIPSCs (bottom) from Ctrl neurons and neurons treated with Oxt 1d or Oxt 3d. (j and k) Average mPSC amplitudes (j) and mPSC frequencies (k; *p<0.05, Ctrl vs. Oxt 1d and Oxt 3d) in Ctrl neurons and neurons treated with Oxt 1d or Oxt 3d. (l and m') Change in EPSC amplitudes (l; Ctrl, n = 49; Oxt 1d, n = 40; Oxt 3d, n = 44) and IPSC amplitudes (m; Ctrl, n = 39; Oxt 1d, n = 35; Oxt 3d, n = 37) during a 10 Hz stimulation train in glutamatergic (l) or GABAergic (m) Ctrl neurons and neurons treated with Oxt 1d or Oxt 3d. Data were normalized to the first response of the respective train. Data are shown as mean ± SEM. Numbers of analyzed cells are indicated in the histogram bars. Statistical analyses were performed with one-way ANOVA followed by post-hoc Bonferroni test. See Table in *Supplementary file 1* for further details.

The following figure supplements are available for figure 2:

**Figure supplement 1.** Oxytocin receptor antagonist atosiban abolishes the oxytocin-induced effects on glutamatergic synaptic transmission.

**Figure supplement 2.** Oxytocin affects early stages of glutamatergic neuron develpment.

**Figure supplement 3.** Oxt exposure affects the development and function of glutamatergic autaptic neurons prepared from E18 mouse embryos.

**Figure supplement 4.** Oxt-induced effects on neuronal function and morphology are mediated by PLCβ.

**Figure supplement 5.** Oxt-induced effects on neuronal function and morphology are mediated by PLCβ.

neurons showed a reduction of their responses to glutamate as compared to untreated cells (*Figure 2f–h*). This indicates that the number of surface glutamate receptors in glutamatergic neurons, but not of $GABA_A$ receptors in GABAergic cells, is decreased by early Oxt exposure, supporting the notion that Oxt specifically affects excitatory neurons. To exert these actions, Oxt requires the Oxtr, since the addition of ATO suppressed the reduction of EPSCs induced by Oxt (*Figure 2— figure supplement 1a–k*) and Oxtr-deficient neurons did not show any changes in their electrophysiological characteristics upon Oxt treatment (*Figure 1—figure supplement 3d–j*). Surprisingly, Oxt effects were restricted to the early phase of synaptic development, as Oxt-application to neurons from DIV 7 onwards, that is, at a time when synaptogenesis is ongoing, did not cause any changes in the morphology and function of glutamatergic neurons (*Figure 2—figure supplement 2a–k*).

In a subset of experiments, we complemented our analyses of cultured neurons from P0 pups, in which Oxt levels reach peak values, with analyses of neurons from E18 mouse embryos. As with P0 cultures (see above), Oxt reduced dendrite complexity and synaptic transmission of glutamatergic neurons in E18 cultures (*Figure 2—figure supplement 3a–j*) without impairing nerve cell survival (*Figure 1—figure supplement 1*). Hence, subsequent experiments were carried out with P0 cultures.

Depending on the context, activated Oxtrs can activate phospholipase Cβ (PLCβ) via $G_{q/11}$, or inhibit or stimulate adenylyl cyclase via $G_{i/o}$ or $G_s$, respectively (*Gimpl and Fahrenholz, 2001*). To test which intracellular pathway mediates the Oxt-induced effects on glutamatergic neurons, we exposed hippocampal neurons to Oxt for 1 DIV in the presence of the PLCβ inhibitor U73122. U73122 blocked all morphological and electrophysiological effects induced by Oxt, whereas U73122 exposure alone was ineffective (*Figure 2—figure supplement 4a–n*). Moreover, Oxt still partially exerted effects on glutamatergic neurons after cultures had been exposed from 1 DIV to 3 DIV to

100 ng/µl pertussis toxin (PTX), which blocks $G_i/G_o$-signaling (*Figure 2—figure supplement 5a–b*). These results indicate that Oxtrs coupled to $G_{q/11}$ modulate hippocampal glutamatergic neurons.

## Inhibitory hippocampal neurons are less affected by Oxt because of low Oxtr expression levels

In contrast to previous reports on acute Oxt effects on GABAergic interneurons (*Mühlethaler et al., 1984*, *1983*; *Owen et al., 2013*; *Zaninetti and Raggenbass, 2000*), our data indicate that Oxt exposure for 1–3 days only affects the morphology of excitatory hippocampal neurons. To further assess this selectivity, we studied autaptic striatal neurons, which are primarily GABAergic. As seen for GABAergic hippocampal neurons, Oxt exposure did not affect functional (*Figure 3a–g*) or morphological parameters (*Figure 3h,k*) of striatal neurons. Further, quantitative analyses of GABAergic pre- and postsynapses, as identified by immunolabeling of VGAT and Gephyrin, respectively (*Nair et al., 2013*), showed that Oxt did not alter the number of GABAergic synapses of striatal neurons (*Figure 3—figure supplement 1a–c*).

To test whether this Oxt insensitivity might be due to a lack of Oxtrs, we overexpressed Oxtrs in striatal cultures using lentiviruses concomitantly encoding Myc-tagged human Oxtr and EGFP. Electrophysiological analyses of Oxtr-overexpressing cells, as identified via EGFP fluorescence, indicated that Oxt treatment for 1 or 3 DIV caused reductions of evoked IPSC amplitudes, of the RRP, and of the response to exogenous application of GABA, which was not seen in control cells expressing EGFP (*Figure 3a–g*). The $P_{vr}$ was not affected (*Figure 3f*). Following early Oxt exposure, Oxtr-overexpressing GABAergic neurons developed less dendritic arborizations than cells expressing only EGFP (*Figure 3i–k*). These data indicate that striatal GABAergic neurons are insensitive to Oxt because they lack Oxtrs, but can be rendered Oxt-sensitive upon ectopic expression of Oxtrs.

To further assess Oxtr expression in hippocampal and striatal GABAergic neurons, we studied Oxtr-Venus knock-in mice ($Oxtr^{Vn/Vn}$) (*Yoshida et al., 2009*), which express Venus instead of the functional Oxtr from the endogenous *Oxtr* locus. Coronal sections from $Oxtr^{Vn/+}$ and $Oxtr^{+/+}$ mice were stained for Venus, and patterns of Venus-expressing neurons, reflecting Oxtr-expressing cells, were evaluated (*Figure 4—figure supplement 1a–b*). As reported (*Yoshida et al., 2009*), we found Venus to be expressed in mouse hippocampus (*Figure 4—figure supplement 1a,c,d*), while no expression was detected in the striatum (*Figure 4—figure supplement 1b*). Venus expression was detectable in the hippocampus at different developmental stages (P3, P5, P10, 3W and 13W; data not shown). We performed a more detailed analysis of the hippocampal Venus expression in three weeks-old $Oxtr^{Vn/+}$ mice, that is, at a time when receptor levels peak (*Hammock and Levitt, 2013*). Venus signal was detected within the entire hippocampus (i.e. rostral, medial, and caudal) and across all subcompartments (e.g. CA1, CA3, and DG). Venus-expressing cells were also positive for the neuron-specific nuclear protein (NeuN) and Map2, indicating their neuronal identity (*Figure 4a,c* and *Figure 4—figure supplement 1c,d*). No Oxtr expression was detected in astrocytes in brain slices immunostained for Venus and the astrocytic marker S100 (*Figure 4—figure supplement 2a,b*), and in contrast to whole brain samples, no *Oxtr* gene expression was detected in primary cultured $Oxtr^{Vn/Vn}$ astrocytes as assessed by Western blotting for Venus (*Figure 4—figure supplement 2c*).

To determine the neuron types expressing Oxtrs in the hippocampus, coronal sections of 3 weeks-old $Oxtr^{Vn/+}$ mice were stained for Venus and the GABAergic neuron marker glutamic acid decarboxylase (GAD) 67 (*Figure 4b*), which is the main GAD isoform in neuronal cell bodies (*Esclapez et al., 1994*; *Solimena et al., 1993*). Quantitative analyses of neurons positive for Venus and GAD67 showed that only 16% of the total GAD-positive neurons in slices express Venus (*Figure 4b,c*). In order avoid a possible underestimation of the number of inhibitory neurons, we repeated this experiment using an antibody against GABA. Quantitative analysis of neurons positive for Oxtr-Venus and GABA showed that only a very small subpopulation of GABAergic cells (19%) express Oxtr (*Figure 4b,d*). Similar results were obtained in hippocampal neuron cultures from $Oxtr^{Vn/+}$ mice (*Figure 4—figure supplement 1e,f*), supporting the notion of a predominant Oxtr expression in hippocampal glutamatergic neurons.

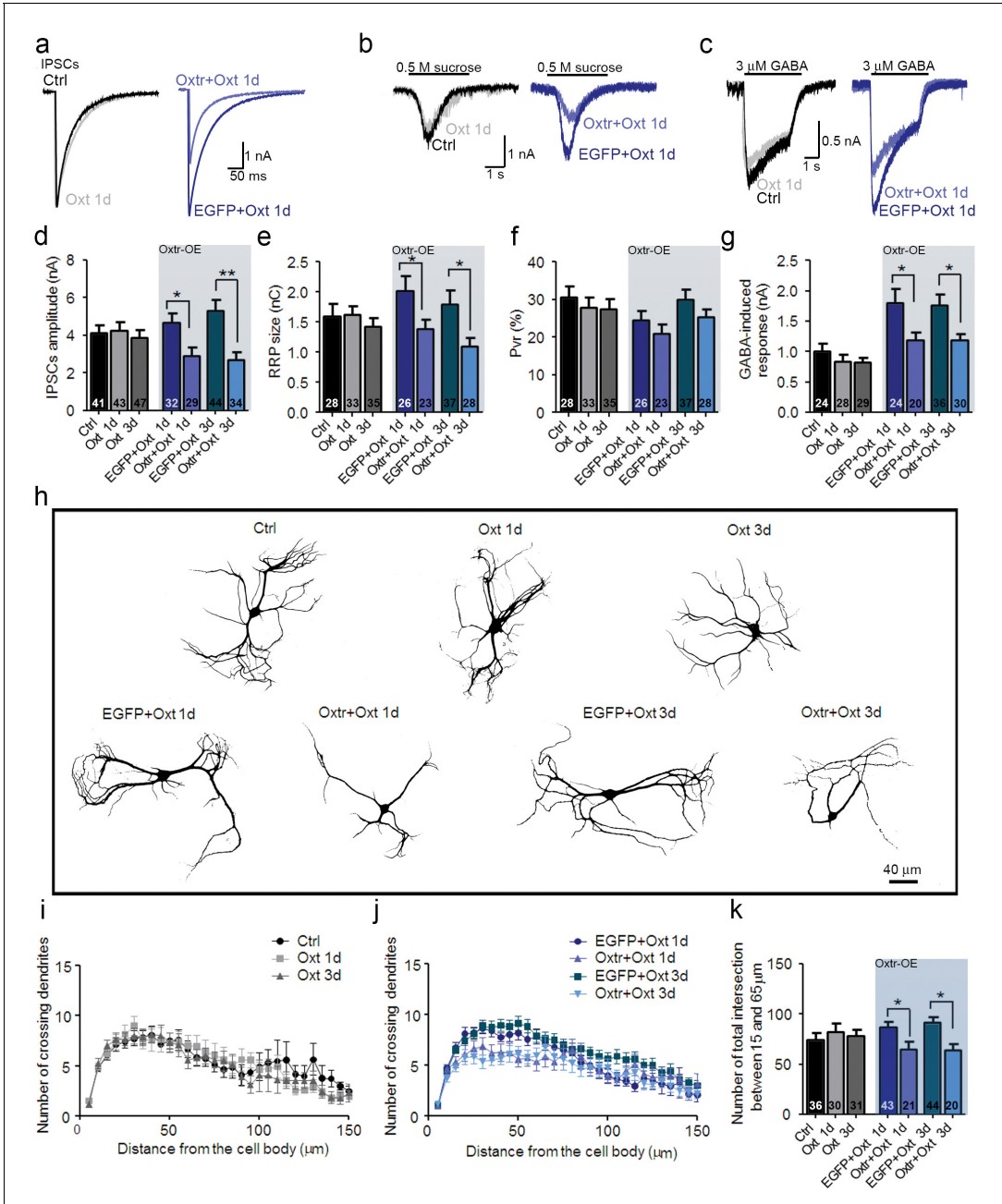

**Figure 3.** Ectopic Oxtr expression confers Oxt sensitivity in striatal inhibitory autaptic neurons. (a–c) Sample traces of evoked IPSCs (a), IPSCs triggered by application of 0.5 M sucrose (b), and IPSCs triggered by 3 μM GABA (c) in an untransfected Ctrl (black), an untransfected cell treated with Oxt for 1 day (Oxt 1d, light grey), an EGFP transfected cell treated with Oxt for 1 day (blue, EGFP+Oxt 1d), and an Oxtr-transfected cell treated with Oxt for 1 day (Oxtr+Oxt 1d, light blue). (d–g) Mean evoked IPSC amplitude (d), RRP size (e), $P_{vr}$ (f), and responses to exogenous 3 μM GABA (g) of untransfected control cell (Ctrl), untransfected cells treated with Oxt for 1 day (Oxt 1d) or 3 days (Oxt 3d), EGFP transfected cells treated with Oxt for 1 day (EGFP +Oxt 1d) or 3 days (EGFP+Oxt 3d), and Oxtr-transfected cells treated with Oxt for 1 day (Oxtr+Oxt 1d) or 3 days (Oxtr+Oxt 3d). The results obtained following lentiviral infection (Oxtr-OE) are highlighted by a light blue background (*p<0.05; **p<0.001 using two-tailed Student's t test). (h) Representative binary images of 14 DIV Map2-labelled striatal neurons. Shown are an untransfected control cell (Ctrl), an untransfected cell treated with Oxt for 1 day (Oxt 1d) or 3 days (Oxt 3d), an EGFP transfected cell treated with Oxt for 1 day (EGFP+Oxt 1d) or 3 days (EGFP+Oxt 3d) and an Oxtr-transfected cell treated with Oxt for 1 day (Oxtr+Oxt 1d) or 3 days (Oxtr+Oxt 3d). (i) Sholl analysis of untransfected control cells (Ctrl) and of untransfected cells treated with Oxt for 1 day (Oxt 1d) or 3 days (Oxt 3d). Number of cells as in panel (k). (j) Sholl analysis of EGFP transfected cells treated with Oxt for 1 day (EGFP+Oxt 1d) or 3 days (EGFP+Oxt 3d), and of Oxtr-transfected cells treated with Oxt for 1 day (Oxtr+Oxt 1d) or 3 days (Oxtr+Oxt 3d). Number of cells as in panel (k). (k) Average numbers of dendrites intersecting Sholl circles at 15–65 μm distance from the cell body in untransfected control cells (Ctrl), untransfected cells treated with Oxt for 1 day (Oxt 1d) or 3 days (Oxt 3d), EGFP transfected cells treated with Oxt for 1

*Figure 3 continued on next page*

*Figure 3 continued*

day (EGFP+Oxt 1d) or 3 days (EGFP+Oxt 3d), and Oxtr-transfected cells treated with Oxt for 1 day (Oxtr+Oxt 1d) or 3 days (Oxtr+Oxt 3d) (*p<0.05, EGFP+Oxt 1d vs. Oxtr+Oxt 1d; *p<0.05, EGFP+Oxt 3d vs. Oxtr+Oxt 3d). Ctrl, control. Data are shown as mean ± SEM. Numbers of analyzed cells are indicated in the histogram bars. Statistical analyses were performed with a two tailed Student's t-test. See *Supplementary files 1* and *2* for further details.

The following figure supplement is available for figure 3:

**Figure supplement 1.** Oxt-treatment does not alter the synapse number of GABAergic autaptic striatal neurons.

## Altered excitation/inhibition balance in cultures of hippocampal neurons lacking Oxtrs

To further study the Oxt effects on the development and function of glutamatergic neurons, we analyzed neurons from $Oxtr^{-/-}$ mice (*Sala et al., 2011*). We first assessed synaptic dysfunctions in mass cultures of hippocampal neurons obtained at embryonic day 18 (E18). Initially, 14 DIV $Oxtr^{-/-}$ and $Oxtr^{+/+}$ neurons were stained using antibodies against VGAT or VGLUT1 and Map2 to label presynapses and dendrites, respectively (*Figure 5a*). Quantitative analyses of VGAT- or VGLUT1-positive fluorescent puncta juxtaposed to Map2-positive dendrites showed that the density of VGAT-positive puncta was reduced and the density of VGLUT1-positive puncta was increased in $Oxtr^{-/-}$ cultures as compared to $Oxtr^{+/+}$ cultures (*Figure 5a,c*). These data complement our finding that the development and synaptogenesis of glutamatergic neurons is perturbed upon exposure to Oxt, and indicate an enhanced excitation/inhibition ratio. This imbalance appears to be caused by altered synapse numbers rather than a change in the ratio between excitatory and inhibitory neurons, because the number of GAD65/67-positive GABAergic neurons was similar in $Oxtr^{-/-}$ and $Oxtr^{+/+}$ cultures (*Figure 5—figure supplement 1a,b*). Likewise, the number of parvalbumin-positive cells, a sub-population of inhibitory interneurons, was similar in hippocampal cultures (*Figure 5—figure supplement 1c,d*) and brain slices (*Figure 5—figure supplement 1e,f*) of $Oxtr^{-/-}$ vs. $Oxtr^{+/+}$ mice. These data indicate that the number of excitatory presynapses is increased in cultures from $Oxtr^{-/-}$ mice whereas the number of inhibitory presynapses is slightly decreased.

To test whether postsynapses are affected by Oxtr deletion, we analyzed the expression of markers of inhibitory postsynapses, that is, Neuroligin 2 (NL2), Gephyrin, and the GABA$_A$ receptor γ2-subunit, and of components of excitatory postsynapses, that is, Neuroligin 1 (NL1), PSD95, and GluA1, in cultured $Oxtr^{-/-}$ and $Oxtr^{+/+}$ neurons. As compared to $Oxtr^{+/+}$ neurons, $Oxtr^{-/-}$ neurons at 14 DIV exhibited an increase of the density of NL1-positive clusters, a decrease of the density of NL2-positive clusters, an increase in the number of excitatory synaptic structures with partially colocalized Synaptophysin and PSD95, and a decrease in the number of inhibitory synaptic structures with partially colocalized Synaptophysin and Gephyrin (*Figure 5d–g*). Further, the number of GluA1-positive clusters was increased in $Oxtr^{-/-}$ cultures, whereas no differences were detected for GABA$_A$γ2 (*Figure 5d–g*). These data again indicate that the ratio between excitatory and inhibitory synapses is shifted towards increased excitation in mass cultures of hippocampal neurons lacking Oxtrs.

To test whether the altered expression of synaptic proteins in $Oxtr^{-/-}$ neurons has any functional impact on synaptic transmission, we analyzed spontaneous EPSCs (sEPSCs) and IPSCs (sIPSCs) in mass cultures of hippocampal $Oxtr^{+/+}$ and $Oxtr^{-/-}$ neurons. The frequency and amplitude of sEPSCs were increased in $Oxtr^{-/-}$ cultures (*Figure 5h,j,k*), while no differences in sIPSC events were detected (*Figure 5i,l,m*). Thus, the increase in the number of apparent glutamatergic synapses, as assessed by immunofluorescence staining, is paralleled by an increase in glutamatergic synaptic activity in $Oxtr^{-/-}$ cultures as compared to $Oxtr^{+/+}$ cultures. Despite slightly lower numbers of presynaptic VGAT-positive structures and of postsynaptic NL2-positive and Gephyrin-positive structures in $Oxtr^{-/-}$ cultures as compared to $Oxtr^{+/+}$ cultures, no significant changes in GABAergic synaptic transmission were detected in $Oxtr^{-/-}$ cultures, which corresponds to the observation that the number of fluorescently labeled clusters of the GABA$_A$γ2 subunit remained unchanged. Probably, the slight reductions of NL2- and Gephyrin-positive puncta in $Oxtr^{-/-}$ cultures are not sufficient to downregulate functional GABA receptors.

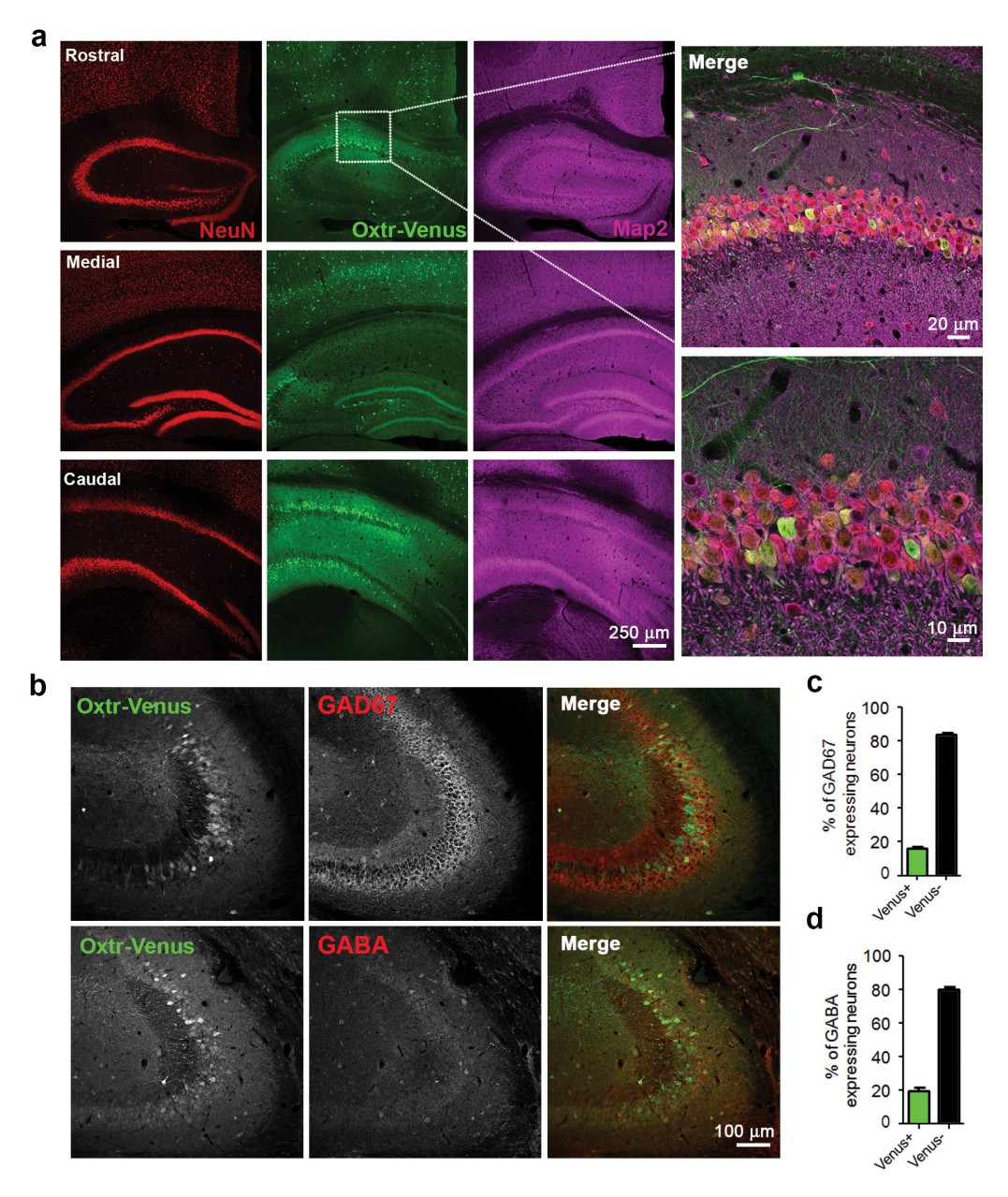

**Figure 4.** Differential and neuron-specific Oxtr expression in mouse hippocampus. (**a**) Immunohistochemical mapping of Venus expression in coronal sections of the rostral, medial, and caudal hippocampus from three weeks old $Oxtr^{Vn/+}$ mice triple-labeled with antibodies against NeuN (red), Venus (green), and Map2 (magenta). (**b**) Confocal images of $Oxtr^{Vn/+}$ hippocampal slices double-labeled for Venus (green) and GAD67 (red) or GABA (red). (**c**) Quantitative analysis of GAD67-positive neurons with (Venus+) and without colabeling for Venus (Venus-) in $Oxtr^{Vn/+}$ hippocampal sections (n = 4–6 sections from two mice). (**d**) Quantitative analysis of GABA-positive neurons with (Venus+) and without colabeling for Venus signals (Venus-) in $Oxtr^{Vn/+}$ hippocampal sections (n = 4–6 sections from three mice). Data are shown as mean ± SEM.

The following figure supplements are available for figure 4:

**Figure supplement 1.** Oxytocin receptors (Oxtr) are expressed in mouse hippocampus.

**Figure supplement 2.** Oxytocin receptors (Oxtr) are not expressed in astrocytes.

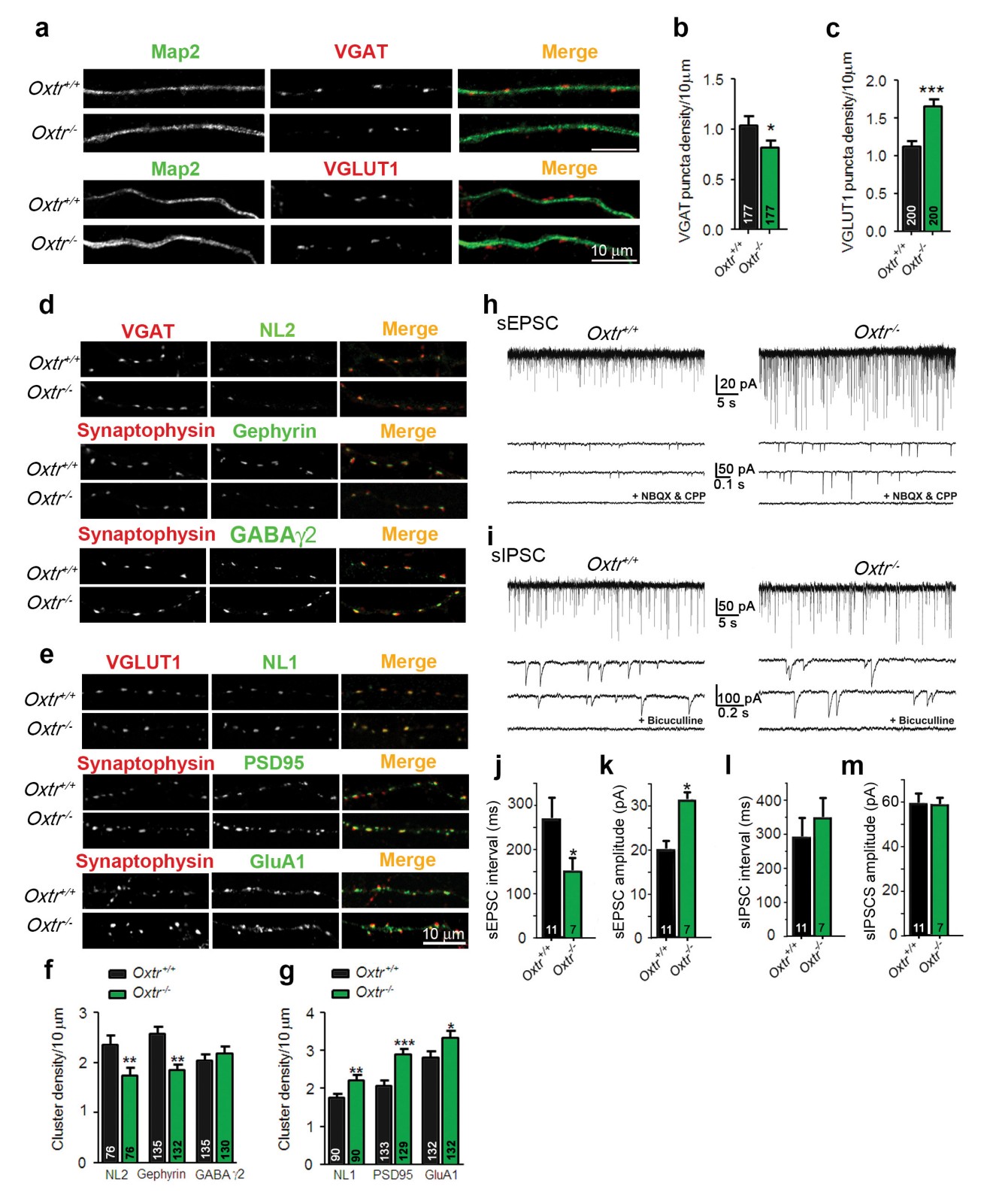

**Figure 5.** Hippocampal cultures from $Oxtr^{-/-}$ mice show an increased excitation/inhibition balance. (**a**) Confocal images of $Oxtr^{+/+}$ and $Oxtr^{-/-}$ 14DIV hippocampal cultures stained for Map2 (green) and VGAT or VGLUT1 (red). (**b** and **c**) Quantitative analysis of the densities of VGAT-positive (**b**) and VGLUT1-positive (**c**) fluorescent puncta juxtaposed to Map2-positive processes, expressed per 10 μm dendrite length (*p<0.05; ***p<0.0001). Numbers of analyzed dendrites are indicated in the histogram bars. (**d**) Immunocytochemical analysis of inhibitory synapses in 14DIV $Oxtr^{-/-}$ and $Oxtr^{+/}$

*Figure 5 continued on next page*

*Figure 5 continued*

[+]hippocampal cultures double-labeled for postsynaptic NL2, Gephyrin, or GABA$_A$ receptor γ2 subunit (green), together with VGAT or synaptophysin (red) to label presynapses. (e) Immunocytochemical analysis of excitatory synapses in 14DIV *Oxtr*[−/−] and *Oxtr*[+/+]hippocampal cultures doubly immunostained for postsynaptic NL1, PSD95, AMPA receptor GluA1-subunit (green), together with VGLUT1 or Synaptophysin (red) to label presynapses. (f and g) Density of puncta positive for the indicated marker proteins of excitatory (f) and inhibitory (g) synapses in *Oxtr*[−/−] and *Oxtr*[+/+] cultures, expressed per per 10 µm dendrite length (*p<0.05; **p<0.001; ***p<0.0001). Numbers of analyzed dendrites are indicated in the histogram bars. (h and i) Sample traces of spontaneous EPSCs (H) and IPSCs (I) in *Oxtr*[+/+] (left) and *Oxtr*[−/−] (right) 11DIV neurons. The bottom traces show recordings in the presence of NBQX (10 µM) and CPP (30 µM) (h), and in the presence of bicuculline (10 µM) (i). (j–m) Mean inter-event intervals and amplitudes of sEPSC (J and K) and sIPSC (L and M) in *Oxtr*[−/−] and *Oxtr*[+/+] neurons (*p<0.05). Data are shown as mean ± SEM. Numbers of analyzed cells are indicated in the histogram bars. Statistical analyses were performed with a two tailed Student's t-test. See *Supplementary file 1* and *2* for further details.

The following figure supplements are available for figure 5:

**Figure supplement 1.** *Oxtr*[+/+] and *Oxtr*[−/−] hippocampal cultures and brain slices show equal numbers of inhibitory GABAergic neurons.

**Figure supplement 2.** Exposure of WT hippocampal cultures to the selective oxytocin receptor agonist TGOT alters the excitatory/inhibitory synapse ratio .

To further assess whether the phenomena seen in autaptic cultures can be corroborated in mass cultures, we exposed primary hippocampal neurons in mass culture to 10 nM of the Oxtr agonist TGOT for 3 DIV following plating. TGOT caused an increase in the number of VGAT- and NL2-positive structures, and a decrease of the number of VGLUT1- and NL1-positive structures (*Figure 5—figure supplement 2a–d*). This was paralleled by a decrease in glutamatergic transmission, as indicated by reduced sEPSC amplitudes and frequencies upon TGOT treatment (*Figure 5—figure supplement 2e–g*). These data again indicate that the early activation of Oxtr in hippocampal neurons shifts the synaptic excitation/inhibition balance towards reduced excitation.

## Oxtr deletion influences hippocampal neuronal morphology and circuit function *in vivo*

Given that a lack of Oxt exposure in utero affects brain development and causes permanent alterations of adult behavior (*Takayanagi et al., 2005*), we investigated if *Oxtr* deletion induces modifications of hippocampal neuronal morphology or circuits *in vivo* that might be related to the behavioral phenotypes exhibited by *Oxtr*[−/−] mice (*Sala et al., 2011*).

We used in utero electroporation to analyze the *in vivo* morphology of hippocampal neurons in P7 *Oxtr*[+/+] and *Oxtr*[Vn/Vn] mice (*Yoshida et al., 2009*). The analysis was initially focused on the CA1 region because in utero electroporation to the ammonic neuroepithelium results in only few targeted CA3 neurons. The dendrite complexity of single CA1 pyramidal cells was visualized by overexpressed membrane-targeted tdTomato. *Oxtr*[Vn/Vn] pyramidal neurons exhibited a more pronounced dendrite complexity at the apical and basal poles as compared to control cells (*Figure 6a*), with an increased number of primary apical dendrite branches and of basal primary neurites, but no changes in the apical shaft length (*Figure 6b–d*). We also detected an increased density of filopodia and filopodia-like protrusions along the apical dendrite shafts of *Oxtr*[Vn/Vn] neurons as compared to *Oxtr*[+/+] cells (*Figure 6e*). These represent the predominant morphological structures found in first postnatal week neurons (*von Bohlen Und Halbach, 2009*). A similarly enhanced dendrite complexity was observed in *Oxtr*[Vn/Vn] pyramidal neurons in CA3 (*Figure 6f,g*). These data indicate that Oxt signaling is required *in vivo* to control the morphology of hippocampal pyramidal cells, in line with the findings on hippocampal neuron cultures that mainly contain pyramidal neurons.

We then performed patch-clamp recordings from CA1 and CA3 pyramidal cells in acute hippocampal slices of juvenile *Oxtr*[+/+] and *Oxtr*[Vn/Vn] mice (P16–P26) to test whether the altered dendrite morphology was accompanied by changes in synaptic transmission. We recorded mEPSCs and mIPSCs and simultaneously filled the patched neurons using 0.4% biocytin for post-hoc morphological studies. We detected an increase of the mEPSC frequency but not of mEPSC amplitudes in *Oxtr*[Vn/Vn] CA1 and CA3 pyramidal neurons (*Figure 7a–c*), while mIPSC events were unchanged (*Figure 7d–f*).

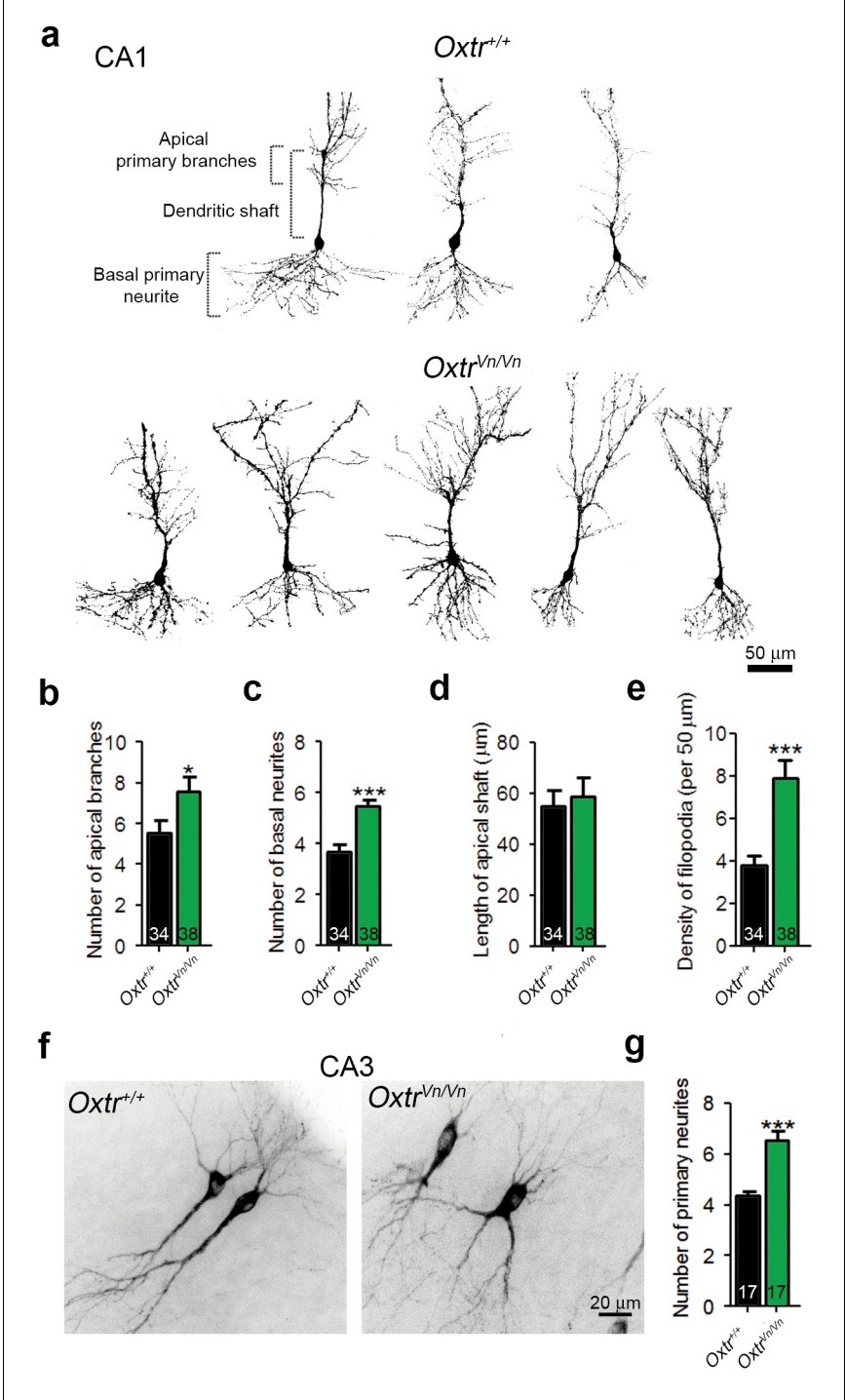

**Figure 6.** Deletion of Oxtrs in hippocampus causes dendrite overgrowth in CA1 and CA3 pyramidal neurons. (**a**) Tracings of CA1 pyramidal neurons in P7 $Oxtr^{+/+}$ and $Oxtr^{Vn/Vn}$ hippocampi expressing a membrane-targeted variant of tdTomato after *in utero* electroporation at E14. (**b** and **c**) Average numbers of primary apical dendrite branches (**B**) and basal primary neurites (**C**) of transfected pyramidal cells in $Oxtr^{+/+}$ and $Oxtr^{Vn/Vn}$ CA1 hippocampus (*p<0.05; ***p<0.0001). (**d**) Mean length of the apical dendrite shaft of transfected CA1 pyramidal cells in $Oxtr^{+/+}$ and $Oxtr^{Vn/Vn}$ hippocampus, measured as a distance between the neuronal soma and the first main bifurcation. (**e**) Average density of filopodia and filopodia-like protrusions along the apical dendrite shaft in transfected CA1 pyramidal cells in $Oxtr^{+/+}$ and $Oxtr^{Vn/Vn}$ hippocampus (***p<0.0001). (**f**) Tracing of CA3 pyramidal neurons in P7 $Oxtr^{+/+}$ and $Oxtr^{Vn/Vn}$ hippocampi expressing a membrane-targeted variant of tdTomato after in utero electroporation at E14. (**g**) Average number of primary dendrites of transfected CA3 pyramidal cells in $Oxtr^{+/}$

*Figure 6 continued on next page*

*Figure 6 continued*

$^{+}$ and $Oxtr^{Vn/Vn}$ hippocampus (**p<0.001 using). Data are shown as mean ± SEM. Numbers of analyzed cells are indicated in the histogram bars. Statistical analyses were performed with a two tailed Student's t-test. See **Supplementary file 1** and **2** for further details.

These results were complemented by the morphological characterization of CA1 and CA3 pyramidal neurons in which mEPSCs and mIPSCs were recorded (*Figure 7a–f*). We detected, as with the *in utero* electroporation experiments, a significant increase in dendritic branching points in CA1 and CA3 $Oxtr^{Vn/Vn}$ neurons (*Figure 8c–j*), indicating an increased dendrite complexity. These changes were more pronounced in CA3 than in CA1 pyramidal neurons and were paralleled by a larger number of total dendritic spines, particularly in basal dendrites (*Figure 8k–m*), where the additional spines were predominantly of the thin and mushroom type (*Figure 8n–o*). No differences in spine density were found in apical and basal dendrites (*Figure 8m*).

The altered morphology of hippocampal pyramidal neurons in $Oxtr^{Vn/Vn}$ mice can impair the function of the corresponding neuronal network, e.g. by disrupting neuronal synchrony (*Kulkarni and Firestein, 2012*; *Yizhar et al., 2011*). Further, changes in synchronous network activity, e.g. in γ-oscillations, have been described in ASD patients (*Rojas and Wilson, 2014*), and in mouse models carrying mutations that cause heritable ASD in humans (*Hammer et al., 2015*). We thus tested whether $Oxtr^{Vn/Vn}$ mice, which due to their lack of Oxtr expression can be regarded as an ASD model, exhibit alterations of γ-oscillations in the hippocampal CA3 region, where γ-oscillations are well characterized and can be measured with higher reliability than in other regions of the hippocampus. We stimulated brain slices with 100 nM kainate, analyzed the oscillatory activity at 25–45 Hz (*Figure 6h*), and found a reduced maximum and average power of γ-oscillations in $Oxtr^{Vn/Vn}$ slices as compared to $Oxtr^{+/+}$ slices (*Figure 6i,j*), without changes in γ-oscillation frequencies. These findings are compatible with the altered balance between synaptic excitation and inhibition in $Oxtr^{Vn/Vn}$ mice and show that a perturbation of Oxt signaling in the hippocampus leads to an aberrant development of pyramidal neurons that, in turn, causes altered γ- oscillations.

## Discussion

Oxytocin regulates brain circuits that control complex mammalian behaviors, including social interactions, and aberrant Oxt signaling has been linked to the pathogenesis of various neurobehavioral disorders, including ASDs (*Feldman et al., 2016*; *Guastella and Hickie, 2016*; *Neumann and Slattery, 2016*). In parturient rodents, the increased Oxt in the bloodstream does not only promote parturition, but also induces changes in the fetal brain, including a switch of GABAergic signaling from excitatory to inhibitory (*Tyzio et al., 2006*). If the Oxt-based communication from the mother to the fetus is perturbed, the offspring develop ASD-like symptoms, indicating that Oxt exerts an early priming influence on brain development that can last far beyond the initial Oxt exposure (*Tyzio et al., 2014*). However, the mechanisms of this Oxt-dependent priming effect that arises prior to and during parturition are unknown. The present study was designed to address this issue by determining whether and how early, transient exposure to Oxt, mimicking the Oxt increase prior to and during parturition, influences neuronal development and function.

### Oxt signaling and dendrite morphogenesis

Our study shows that transient Oxt exposure of cultured mouse hippocampal glutamatergic neurons causes altered neuronal dendrite complexity and altered numbers of excitatory synapses. These Oxtr- and $G_{q/11}$-PLCβ-mediated effects (*Figures 1* and *2* and *Figure 2—figure supplements 1* and *4*) were only observed upon Oxt treatment immediately after seeding, that is, at an early stage of neuronal development, whereas later Oxt addition was ineffective (*Figure 2—figure supplement 2a–k*). This indicates that Oxt exerts a 'priming' effect on dendritogenesis and synaptogenesis and is not required continuously or for the consolidation of preexisting dendrites or synapses. Accordingly, Oxtr loss in hippocampal pyramidal neurons leads to an increase in dendrite complexity in vivo (*Figure 6a–e*).

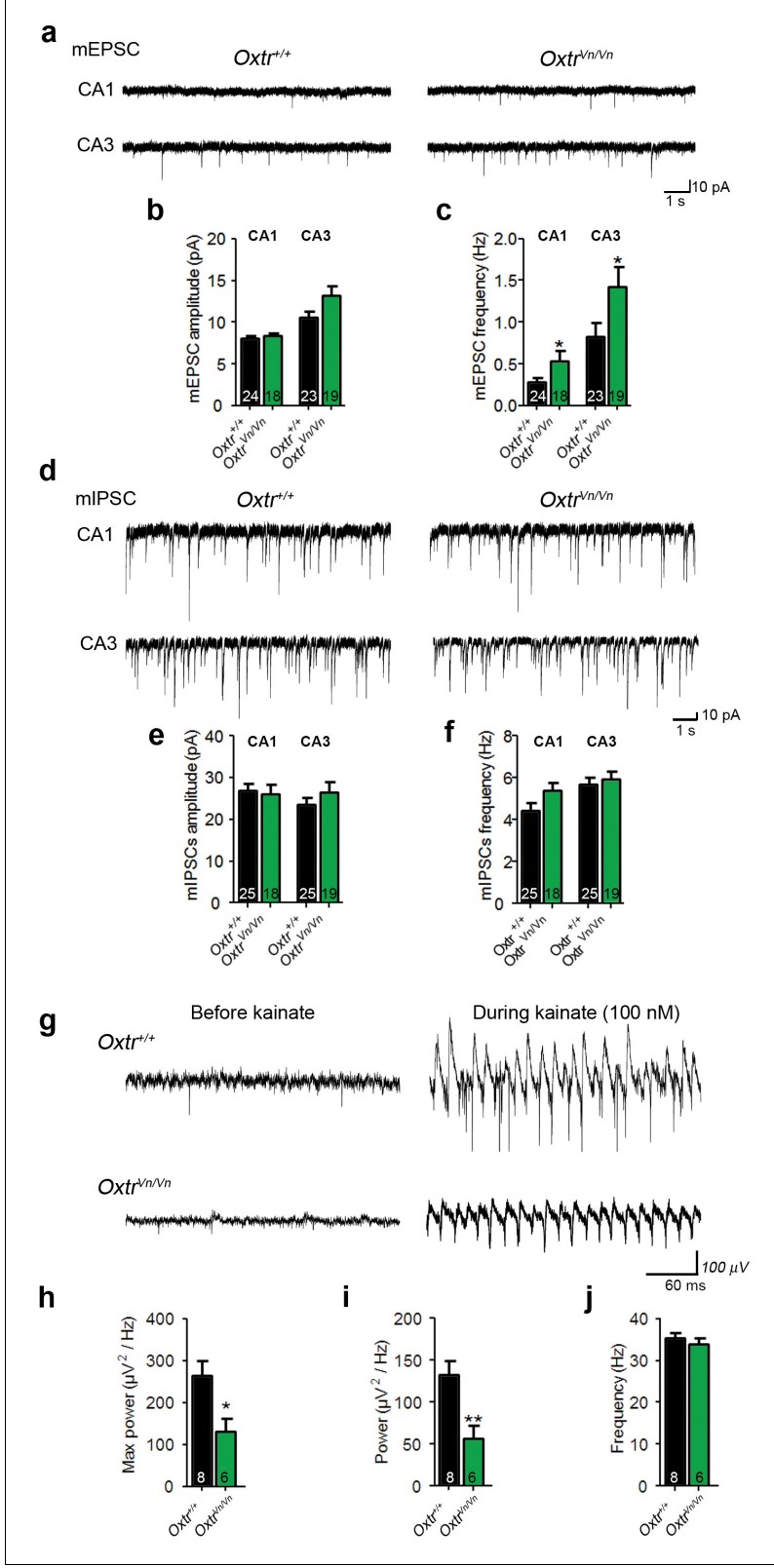

**Figure 7.** Loss of Oxtr increased excitatory synaptic transmmission and perturbed γ-oscillation in hippocampus. (a) Representative traces of mEPSCs recorded from CA1 (top) and CA3 (bottom) pyramidal neurons in *Oxtr*[+/+] (right) and *Oxtr*[Vn/Vn] (left) hippocampal slices. (b, c) Average mEPSC amplitude (B) and frequency (C) in CA1 and CA3 pyramidal neurons from *Oxtr*[+/+] and *Oxtr*[Vn/Vn] mice (*p<0.05). (d) Representative traces of mIPSCs recorded from

*Figure 7 continued on next page*

*Figure 7 continued*

CA1 (top) and CA3 (bottom) pyramidal neurons in $Oxtr^{+/+}$ (right) and $Oxtr^{Vn/Vn}$ (left) hippocampal slices. (e–f) Average mIPSC amplitude (B) and frequency (C) in CA1 and CA3 pyramidal neurons from $Oxtr^{+/+}$ and $Oxtr^{Vn/Vn}$ mice. (g) Representative recordings of γ-oscillations in the CA3 region of hippocampal slices from $Oxtr^{+/+}$ (top) and $Oxtr^{Vn/Vn}$ mice (bottom) before (left, baseline) and during (right) 100 nM kainate application. (h–j) Quantification of the maximum (h) and average (i) power and of the frequency (j) of kainate-induced (100 nM) γ-oscillations oscillations recorded in $Oxtr^{+/+}$ and $Oxtr^{Vn/Vn}$ hippocampal slices (*p<0.05; **p<0.001). Data are shown as mean ± SEM. Numbers of analyzed cells (b–f) and mice (h–j) are indicated in the histogram bars. Statistical analyses were performed with a two tailed Student's t-test. See *Supplementary files 1* and *2* for further details.

Our data provide the first evidence for a role of Oxt in the control of dendrite growth and synaptogenesis in glutamatergic pyramidal neurons of the hippocampus, a brain structure that receives Oxt signaling via direct Oxt neuron projections and by Oxt diffusion after somato-dendritic release. In contrast, previously reported effects of Oxt on neuronal morphology mostly concerned autocrine Oxt signaling, which can have different effects depending on the context. For instance, Oxt released from the somato-dendritic compartments of magnocellular neurons in the SON is thought to stimulate dendrite development in an autocrine fashion (*Chevaleyre et al., 2001*) likely in synergy with afferent glutamate release (*Chevaleyre et al., 2002*), whereas during lactation, when Oxt levels are high, dendrite branching decreases in magnocellular SON neurons (*Stern and Armstrong, 1998*).

Downstream effectors involved in the Oxt-dependent regulation of hippocampal neuron morphology remain unknown, but transient Oxt signaling likely exerts its morphoregulatory effects by controlling gene transcription programs that regulate various components of growth-promoting pathways or the cytoskeleton. In neuroblastoma and glioblastoma cells, for instance, Oxtr agonists differentially regulate the genes encoding Nestin and Map2 (*Bakos et al., 2013*). In addition, activation of the PLCβ pathway, which mediates the effects of Otx on dendrite development in hippocampal neurons (*Figure 2—figure supplement 1*), affects dendrite morphology. For example, tonic activation of $α_1$-adrenergic receptors activates the PLCβ pathway to cause dendrite retraction and reduced spine density in pyramidal neurons (*Cook and Wellman, 2004*; *Radley et al., 2006*, *2008*; *Wellman, 2001*), and sustained PKC activation downstream of PLCβ signalling modulates spine density in hippocampal neurons by perturbing actin cytoskeleton cross-linking (*Calabrese and Halpain, 2005*; *Hains et al., 2009*). Further, PKC activity is known to affect dendrite development in several other neuronal cell types. For instance, PKC activators cause a strong reduction of the dendrite complexity in Purkinje cells, while PKC inhibition leads to increased dendrite size (*Metzger and Kapfhammer, 2000*). Hence, it is likely that the concerted action of such pathways downstream of Oxtr activation ultimately converge to control neuron growth and synaptogenesis as described here.

## Cell type specificity of morphoregulatory Oxt effects

A striking and unexpected finding of our study is that the morphoregulatory effects of Oxt appear to specifically target glutamatergic but not GABAergic neurons. Our analysis shows that this specificity is caused by the fact that GABAergic neurons express very low Oxtr levels or none at all (*Figures 1*, *2* and *4* and *Figure 4—figure supplement 1*). Immunohistochemical mapping of Oxtr-expressing neurons in brain sections of $Oxtr^{Vn/+}$ mice showed that Oxtrs are widely expressed throughout the entire hippocampus and that all hippocampal subregions contain Oxtr expressing cells (*Figure 4* and *Figure 4—figure supplement 1*). This is in contrast to a previous report indicating that the Oxtr is mainly expressed in the CA3 subregion (*Hammock and Levitt, 2013*) but nicely matches recent observations obtained using a novel specific antibody to the mouse Oxtr (*Mitre et al., 2016*), showing Oxtr expression in the pyramidal cell layer in all hippocampal subfields. More importantly, the majority of Oxtr-expressing neurons are GAD-67-negative (*Figure 4* and *Figure 4—figure supplement 1*). Further, Oxtr expression is undetectable in the striatum, which is mostly composed of GABAergic neurons, and Oxtr expression in cultured striatal neurons rendered these cells sensitive to the morphoregulatory effects of Oxt (*Figure 3*). In essence, these findings show that GABAergic neurons contain the intracellular signaling pathways that mediate the Oxt effects on neuronal morphology but are intrinsically insensitive to Oxt due to the lack of Oxtrs.

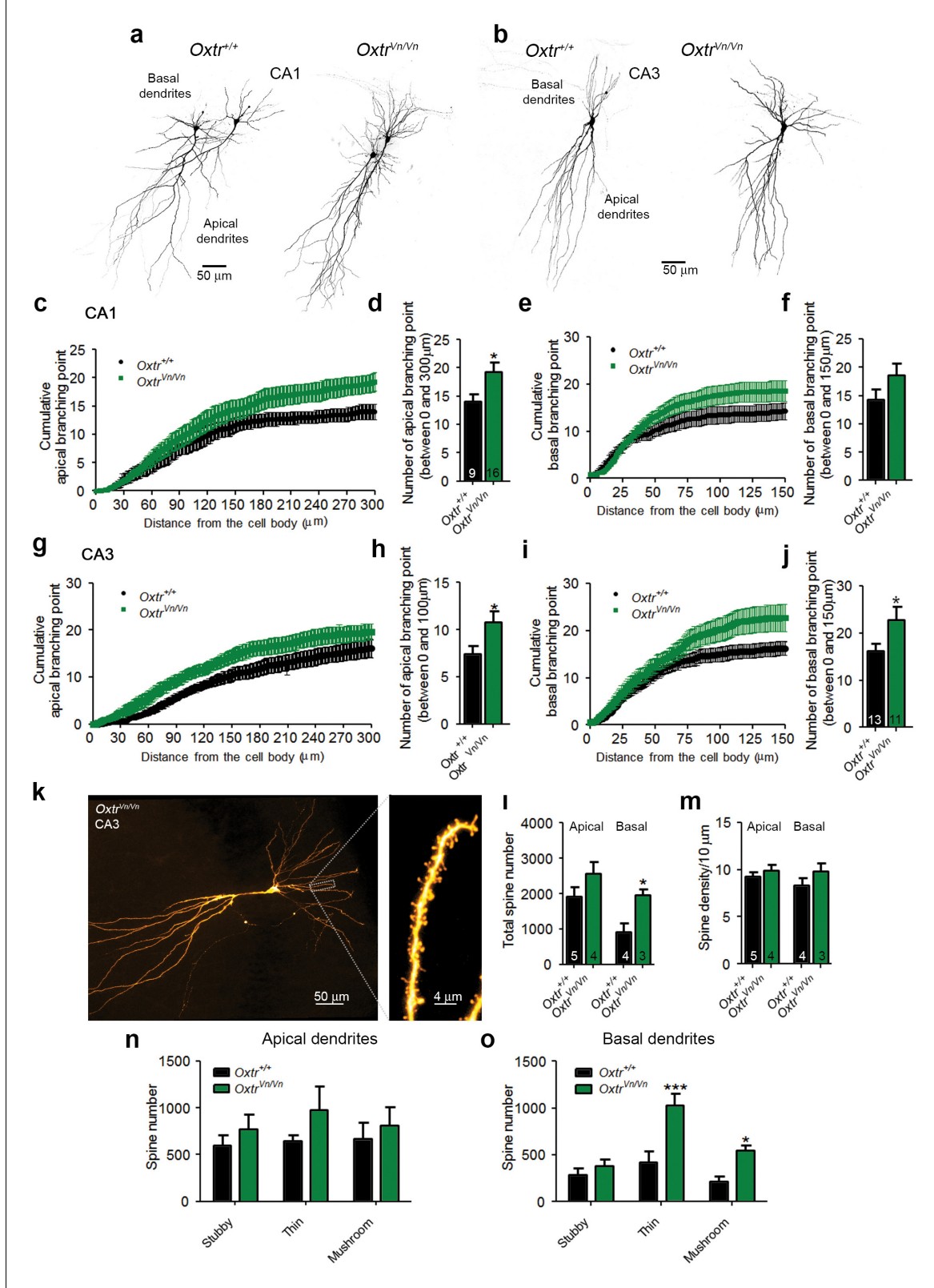

**Figure 8.** Altered dendritic complexity and spine number in Oxtr-deficient pyramidal hippocampal neurons. (**a, b**) Representative images of biocytin-filled CA1 (**a**) and CA3 (**b**) pyramidal neurons in *Oxtr*[+/+] and *Oxtr*[Vn/Vn] hippocampal slices. (**c, d**) Quantification of the cumulative number of apical branching points (**c**) and of the total number of apical branching points (**d**) in dendrites of CA1 pyramidal neurons from *Oxtr*[+/+] and *Oxtr*[Vn/Vn] hippocampal slices (*p<0.05). (**e, f**) Quantification of the cumulative number of basal branching points (**e**) and of the total number of basal branching

*Figure 8 continued on next page*

*Figure 8 continued*

points (f) in dendrites of CA1 pyramidal neurons from $Oxtr^{+/+}$ and $Oxtr^{Vn/Vn}$ hippocampal slices. (g, h) Quantification of the cumulative number of apical branching points (g) and of the total number of apical branching points (h) in dendrites of CA3 pyramidal neurons from $Oxtr^{+/+}$ and $Oxtr^{Vn/Vn}$ hippocampal slices (*p<0.05). (i, j) Quantification of the cumulative number of basal branching points (i) and of the total number of basal branching points (j) in dendrites of CA1 pyramidal neurons from $Oxtr^{+/+}$ and $Oxtr^{Vn/Vn}$ hippocampal slices (*p<0.05). (k) Confocal image of a biocytin-filled CA3 pyramidal neuron. (l, m) Quantification of the total spine number (l) and average spine density (m) in CA3 pyramidal neurons from $Oxtr^{+/+}$ and $Oxtr^{Vn/Vn}$ hippocampal slices (*p<0.05). (n, o) Spine classification in apical (n) and basal (o) dendrites of CA3 pyramidal neurons from $Oxtr^{+/+}$ and $Oxtr^{Vn/Vn}$ hippocampal slices (*p<0.05; ***p<0.0001). Data are shown as mean ± SEM. Numbers of analyzed cells are indicated in the histogram bars. Statistical analyses were performed with a two tailed Student's t-test. See *Supplementary file 1* and *2* for further details.

Our finding that Oxt exerts its morphoregulatory effects selectively on excitatory neurons is unexpected in the light of several publications reporting that, in rat, Oxt acutely activates interneurons located in the stratum pyramidale and modulates fast-spiking interneurons, thus enhancing inhibitory synaptic transmission in the hippocampus (*Mühlethaler et al., 1984*, *1983*; *Owen et al., 2013*; *Zaninetti and Raggenbass, 2000*). One explanation for these discrepancies could be a species difference between mouse and rat as regards Oxt sensitivity of inhibitory neurons. Indeed, species-specific effects of Oxt in rat and guinea pig hippocampal neurons were described previously (*Raggenbass et al., 1989*). A second explanation concerns the respective experimental design. We aimed at analyzing the effects of an extended but transient Oxt exposure of isolated neurons to mimic the diffusive action of Oxt in the fetal brain prior to and during parturition, while previous studies mainly addressed acute effects of Oxt in hippocampal slices. Such acute effects may even include alterations of inwardly rectifying $K^+$ channel conductance (*Gravati et al., 2010*). In an acute Oxt treatment scenario in slices, indirect circuit effects of Oxt signaling cannot be excluded, and priming effects of Oxt signaling and their long-term consequences for neuronal morphology and function cannot be detected. A third explanation relates to Oxtr and Avpr pharmacology. Most previous studies employed concentrations of Oxtr ligands that are likely to also activate Avprs, so that Oxtr independent effects cannot be excluded. In contrast, the Oxt, Oxtr agonist, and Oxtr antagonist concentrations used in the present study were chosen based on the relative Oxt affinities of mouse Oxtrs and Avprs in order to ensure selective targeting of Oxtrs (*Busnelli et al., 2013*).

## Oxt treatment vs. *Oxtr* KO

Our data on the morphology and function of Oxt-treated glutamatergic autaptic neurons from WT hippocampus and on the morphology of pyramidal hippocampal neurons in brains of Oxtr-deficient $Oxtr^{Vn/Vn}$ mice are reverse complements. In the former, we observed reduced dendrite complexity and reductions in synapse numbers and function (*Figures 1* and *2*), while in the latter we observed an increased dendritic complexity in vivo in P7 animals (*Figure 6*) and in acute hippocampal slices obtained from juvenile mice, along with corresponding physiological changes (*Figures 7* and *8*). These findings are compatible with the notion of a priming role of Oxt/Oxtr receptor signaling in early phases of neuronal development.

Likewise, our data on hippocampal neuron mass cultures from $Oxtr^{-/-}$ mice are essentially in agreement with the data obtained with Oxt-treatment of autaptic cultures and with slices from $Oxtr^{Vn/Vn}$ mice, but the details are more complex. Because the neurons were cultured from embryos after hysterectomy at gestation day 18 and because the culture medium used in our experiments most likely does not contain Oxt, phenotypic changes seen in cultured $Oxtr^{-/-}$ neurons likely arise from a lack of Oxtr signaling late in gestation, when Oxt levels are already increased (*Pfaff et al., 2002*). We detected an increased number of excitatory synapses in $Oxtr^{-/-}$ cultures as compared to $Oxtr^{+/+}$ cultures, along with an increase of sEPSC frequency and amplitude (*Figure 5*). This parallels the changes in hippocampi of $Oxtr^{Vn/Vn}$ mice, in which glutamatergic neurons showed more complex dendrites (*Figure 6*), and the increased excitatory neurotransmission and spine number in acute hippocampal slices (*Figure 8*). Moreover these observations reversely complement the data on Oxt treated autaptic glutamatergic neurons from WT mice, which showed reductions in dendrite complexity, synapse number, and synapse function (*Figures 1* and *2*).

In addition, however, the mass culture experiments revealed a negative effect of Oxtr deletion on the number of protein clusters containing the inhibitory synapse components NL2 and Gephyrin (but

not GABA$_A$γ2) (*Figure 5a–g*). No corresponding opposite effect was seen upon Oxt treatment of autaptic GABAergic neurons (*Figure 2*, *Figure 1—figure supplement 1* and *Figure 3—figure supplement 1*). It is possible that this discrepancy is due to homeostatic mechanisms that control the interplay between excitation and inhibition in mass cultures but are irrelevant in single-neuron autaptic cultures. Apart from this, it is possible that the absolute number of inhibitory synapses in mass culture is not changed but that synapses are simply 'diluted' over the larger dendrites of glutamatergic $Oxtr^{-/-}$ neurons, which represent about 80% of the cells in the culture. This could be the cause of the small reduction in the density of inhibitory synapses as detected by NL2 and Gephyrin labeling, which might escape detection by GABA$_A$γ2 labeling. This notion is supported by the fact that sIPSC frequency is unchanged in mass-cultured $Oxtr^{-/-}$ neurons (*Figure 5I*).

## Oxt signaling and neurodevelopmental disorders

Many studies have linked dysfunctions of Oxt signaling to ASDs and other neuropsychiatric disorders. The present data indicate that one consequence of pathologically low Oxt signaling, as mimicked in the $Oxtr^{Vn/Vn}$ hippocampus (*Figures 6*, *7* and *8*), is dendrite overgrowth in glutamatergic neurons of the hippocampus, and possibly also in other brain regions. Interestingly in this context, defects in neuronal dendrite organization feature in multiple neurodevelopmental disorders, including ASDs (*Kulkarni and Firestein, 2012*). Dendrite hypertrophy akin to the consequence of deficient Oxt signaling described here was observed in the PTEN KO and Fmrp KO mouse modes of ASD (*Galvez et al., 2003*; *Kwon et al., 2006*; *Restivo et al., 2005*; *Williams et al., 2015*), and currently available morphometric data on ASD patients, although heterogeneous, indicate dendrite and brain volume changes in some neuron populations and brain regions that are compatible with ASD-related dendrite hypertrophy (*Bennett and Lagopoulos, 2015*; *Courchesne et al., 2003*; *Williams and Casanova, 2011*), including evidence for an enlarged hippocampus (*Schumann et al., 2004*).

The increased dendrite branching of excitatory neurons caused by deficient Oxt signaling likely leads to an increase in the balance between excitation and inhibition in neuronal networks (*Figures 7* and *8*), which might also ultimately be the cause of the increased seizure susceptibility of $Oxtr$-deficient mice (*Sala et al., 2011*). Such a disturbance in the excitation/inhibition balance has repeatedly been invoked as a causal contributor to ASD pathology (*Gillberg and Billstedt, 2000*; *Rubenstein and Merzenich, 2003*). Indeed, most genes known to be involved in the genetic etiology of ASDs encode proteins that directly or indirectly determine synapse density or function and hence excitation/inhibition balance (*Kleijer et al., 2014*), and, correspondingly, excitation/inhibition imbalances are a key feature of numerous genetic mouse models of ASD (*Gogolla et al., 2009*). The present study shows that $Oxtr$-deficient mice belong into this category of ASD models, stressing the notion that an altered excitation/inhibition balance is a key feature of ASDs.

The aberrant morphology of pyramidal cells in the hippocampus of $Oxtr^{Vn/Vn}$ mice is paralleled by a strong reduction in the power of γ-oscillations (*Figure 7g–j*). These oscillations are thought to play a key role in cognitive functions mediated by the hippocampus, such as memory coding, memory retrieval, and working memory (*Colgin and Moser, 2010*), and their perturbation has been linked to multiple brain disease states. Similar alterations in CA3 γ-oscillations, although likely caused by a different cellular mechanism, were observed in other genetic mouse models of ASD (*Hammer et al., 2015*), and perturbed γ-oscillatory activity can be detected in ASD patients (*Rojas and Wilson, 2014*) and other neuropsychiatric disorders (*Maxwell et al., 2015*; *Woo et al., 2010*). These findings led to the idea that perturbations of γ-oscillations might serve as a 'biomarker' for the corresponding disorders, which is supported by the present data on $Oxtr$-deficient mice. Interestingly, the specific activation of GABAergic cells during acute application of the Oxtr agonist TGOT in rat hippocampal slices increases the population spike during kainate-induced γ-oscillations, without inducing any changes in oscillation power (*Owen et al., 2013*). While it is not clear whether these effects are due to a specific role of Oxt-signaling in the rat, which is not relevant in the mouse, the corresponding data show that short-term action of Oxt is not sufficient to induce changes in neuronal network function although excitation/inhibition balance is changed transiently. In general terms, the perturbation of γ-oscillations in $Oxtr^{Vn/Vn}$ mice may be a consequence of the altered excitation/inhibition balance during the early stages of neuronal network formation. In all current models of γ-oscillatory activity, inhibitory interneurons, such as basket cells targeting the somata of principal cells, play a key role (*Bartos et al., 2007*; *Buzsáki and Wang, 2012*). It is possible that more complex dendrites of CA3 principal cells in $Oxtr^{Vn/Vn}$ mice receive more excitatory inputs as shown by the increased number of

spines observed in CA3 pyramidal neurons, causing an increased excitatory drive onto these cells that partly overrides the inhibitory inputs. An alternative scenario would be that the increase in dendrite complexity of CA3 neurons leads to a reduced input resistance of these cells, a consequent reduction in their activity, and a reduction in γ-oscillations.

## Conclusion

Our results indicate that Oxt signaling exerts an early priming effect on developing glutamatergic neurons in the hippocampus. This process throttles dendrite development and thus guarantees a proper excitation/inhibition balance. Aberrations in this process, caused for instance by genetic perturbation of Oxt signaling, as shown here (*Figure 7*), or by deficient Oxt signaling during pregnancy, parturition, or early postnatal brain development, cause dendrite hypertrophy, an increased excitation/inhibition balance, and consequent defects in network synchrony in the hippocampus, and likely also in other brain regions. These findings provide important insights into the role of Oxt signaling in fetal and early postnatal brain development, and represent an explanatory basis for the etiological role that dysfunctions in Oxt signaling appear to play in ASDs and other neuropsychiatric disorders.

# Materials and methods

## Animals

$Oxtr^{-/-}$ (*Takayanagi et al., 2005*) and $Oxtr^{Vn/+}$ mice (*Yoshida et al., 2009*) were provided by K. Nishimori (Tohoku University, Miyagi, Japan). $Oxtr^{-/-}$ animals were maintained and genotyped as described (*Sala et al., 2011*) while $Oxtr^{Vn/+}$ mice were routinely genotyped by PCR using genomic DNA extracted from tail biopsies. For amplification the following oligonucleotide primers were used: forward (5'-AATTCCAGAGTGTCTCGTGTGGC-3'); reverse (5'-TGCACCGTCTAGGGTTAGAACAGA −3'; 5'-CGCATCGCCTTCTATCGCCTTCTT −3). The PCR reaction was carried out for 32 cycles with denaturation at 94°C for 30 s, annealing at 62°C for 1 min, and extension at 72°C for 1 min. $Oxtr^{+/+}$ and $Oxtr^{Vn/Vn}$ genotypes were discriminated by the presence of specific 339 bp and 445 bp PCR products, respectively. C57BL/6J mice employed for the experiments with TGOT were purchased from Harlan Laboratories or bred at the Max Planck Institute of Experimental Medicine from mice obtained from Charles River Laboratories. All animal experiments were performed in accordance with the guidelines for the welfare of experimental animals issued by the State Government of Lower Saxony, Germany, and the Italian Government, in compliance with European and NIH guidelines (33.9-42502-04-13/1359; 33.9-42502-04-13/1052).

## Mass culture of hippocampal neurons and autaptic neuronal culture

Microisland cultures of hippocampal and striatal neurons were prepared and cultured as previously reported (*Burgalossi et al., 2012*; *Jockusch et al., 2007*). Briefly, astrocytes for autaptic cultures were obtained from mouse cortices dissected from P0 WT animals and enzymatically digested for 15 min at 37°C with 0.25% (w/v) trypsin-EDTA (Gibco). Astrocytes were plated in T75 culture flasks in DMEM (Gibco) containing 10% FBS and penicillin (100 U/ml)/streptomycin (100 μg/ml), and grown for 7–10 days in vitro (DIV). After this, astrocytes were trypsinized and plated at a density of ~30,000 cells/well onto 32 mm-diameter glass coverslips that were previously coated with agarose (Sigma-Aldrich), and stamped using a custom-made stamp to generate 200 μm × 200 μm substrate islands with a coating solution containing poly-D-lysine (Sigma-Aldrich), acetic acid, and collagen (BD Biosciences). Hippocampi and striata from P0 WT or $Oxtr^{Vn/Vn}$ mice and from E18 mouse embryos were isolated and digested for 60 min at 37°C in DMEM containing 2.5 U/ml papain (Worthington Biomedical Corporation), 0.2 mg/ml cysteine (Sigma), 1 mM CaCl2, and 0.5 mM EDTA. After washing, the dissociated neurons were seeded onto the microisland plates in pre-warmed Neurobasal medium (Gibco) supplemented with B27 (Gibco), Glutamax (Gibco) and penicillin (100 U/ml)/streptomycin (100 μg/ml) at a density of ~4000 cell/well. Autaptic cultures were exposed to Oxt (100 nM; Bachem), atosiban (100 nM; Sigma-Aldrich), or both during the first 3 DIV by adding fresh drug solution every 24 hr. Oxt concentration was chosen based on the pharmacological features of the relevant receptors (*Busnelli et al., 2013*), that is, the murine Oxtr ($K_d$ ~1 nM for Oxt) and vasopressin V1a receptor ($K_d$ ~20–30 nM for Oxt). The other compound used in the present study, 1-[6- ((17$\beta$−3-methoxyestra-1,3,5 (10)-trien-17-yl)amino)hexyl]−1H-pyrrole-2,5-dione (U-73122, 3 μM; Calbiochem),

was applied to cells alone or in combination with Oxt (100 nM) only at DIV one due to the excessive neuronal death observed after three-day exposure. Striatal autaptic cultures were infected 6 hr after plating and treated with Oxt starting from 1 to 3 DIV. Electrophysiological recordings were performed between 9 and 14 DIV, whereas immunofluorescence analyses were carried out at 14 DIV. In order to contain the intrinsic variability of the autaptic cultures preparation, we employed no less than four different neuronal preparations for every condition analyzed, typically studied numbers high of cells and focused all analyses on DIV10–14. Further, all experiments were conducted by measuring on every day during the experiments approximately the same number of cells per condition. Finally, within-experiment controls were included in all the experiments. For cell death quantification (*Figure 1—figure supplement 1*), WT mass cultures were plated onto 25 mm poly-L-lysine (Sigma-Aldrich) coated coverslips at a density of ~90,000 cells/well. $Oxtr^{Vn/Vn}$ high density cultures (*Figure 4—figure supplement 1f*) were prepared following the same protocol but were plated onto 18 mm poly-L-lysine (Sigma-Aldrich) coated coverslips at a density of ~25,000 cells/well. $Oxtr^{+/+}$ and $Oxtr^{-/-}$ primary mass hippocampal cultures (*Figure 5*) were generated as described (*Kaech and Banker, 2006*). Due to the aberrant maternal behaviour and the defective lactation of $Oxtr^{-/-}$ dams, which results in the premature death of offspring (*Takayanagi et al., 2005*), $Oxtr^{-/-}$ hippocampal neurons were routinely prepared from embryos at E18. Briefly, hippocampi of E18 mouse embryos were isolated and dissociated by treatment with 2.5 g/L trypsin for 15 min at 37°C and repeated pipetting through a fire-polished Pasteur pipette. The cells were then plated onto 24 mm poly-L-lysine (Sigma-Aldrich) coated glass coverslips at a density of ~160,000 cells/well. For each preparation, two pregnant mice of each genotype type carrying 6–8 embryos each were used. Neurons were grown in B27-supplemented (Invitrogen-Gibco) Neurobasal medium, and maintained at 37°C and 5% $CO_2$ for up to 14 DIV. Primary hippocampal mass cultures from C57BL/6J mouse hippocampi (*Figure 5—figure supplement 2*) were exposed to the selective Oxtr agonist Thr4-Gly7-Oxytocin (TGOT, 10 nM; Bachem) during the first 3 DIV after plating by adding fresh solution every 24 hr.

## DNA constructs

Expression vectors encoding lentiviral supplementary proteins (pCMVdeltaR8.2 and VSV-G) and the pFUGW expression vector were described previously (*Hsia et al., 2014*). The plasmid encoding the membrane-targeted mutant of tdTomato (tdTom BsrGI) used for the *in utero* electroporation experiments was provided by V. Tarabykin (Charité, Berlin, Germany). The f(syn)-ugw-rbn vector was donated by C. Rosemund (Charité, Berlin, Germany).

## Lentivirus production

Lentiviruses were generated as described previously with only slight modifications of the original protocol (*Naldini et al., 1996*). HEK293FT cells (Gibco; R700–07; RRID:CVCL_6911) plated on a poly-L-lysine (Sigma-Aldrich) coated 15 cm plastic dish were transfected with the packaging pVSV-G, the envelope pCMVdeltaR8.2, and the backbone vectors with Lipofectamine 2000 (Invitrogen), according to the manufacturer's instructions. Cells were incubated with the transfection mix for 6 hr in Opti-MEM medium (Gibco) containing 10% fetal bovine serum (FBS, Gibco), and the medium was then changed to Dulbecco's Modified Eagle's Medium (DMEM, Gibco) containing 2% FBS, 10 units/mL penicillin (Gibco), 10 µg/mL streptomycin (Gibco), and 10 mM sodium butyrate (Merck). Forty-eight hours after transfection, the culture medium containing the lentiviral particles was harvested and concentrated using Amicon particle centrifugal filters (100 kDa; Millipore) according to the manufacturer's instructions. High-titer lentiviral samples were aliquoted, snap-frozen in liquid nitrogen, and stored at −80°C until used. HEK293FT cells, which were only used for virus production, were obtained from a low passage culture (P3) from the original cell line purchased from Gibco and checked for mycoplasma every 6 months.

## Western blotting

Western blotting samples from 2-weeks old $Oxtr^{Vn/+}$ brains and from 7-days old primary astrocyte cultures obtained from $Oxtr^{Vn/+}$ interbreedings were prepared as described (*Hammer et al., 2015*). After SDS-PAGE and blotting, correct total protein loading (15 µg per lane) was validated by MEM-Code (Pierce, Thermo Scientific). Nitrocellulose membranes were probed with antibodies to Venus

(mouse anti-GFP, Roche, clone 7.1/13.1, 1:4000; AB_390913), GFAP (guinea pig anti-GFAP, Synaptic Systems, 173004, 1:1000; RRID:AB_10641162), or actin (mouse anti-actin, Sigma, AC40, 1:2000, RRID:AB_476730) and corresponding horseradish peroxidase (HRP) conjugated goat antibodies (Jackson ImmunoResearch Laboratories, 1:5000–1:20,000).

## Assessment of cell death

To assess if Oxt-treatment induces apoptosis, we quantified the percentage of pyknotic nuclei in hippocampal E18 and P0 cultures. For the nuclear condensation analysis, cultures were fixed at 12 DIV with 4% PFA and nuclei were stained following an incubation for 20 min with 1 μM DAPI in PBS. Images were obtained using a Carl Zeiss Apoptome (Zeiss Axio Imager Z1) microscope. All images were processed equally (binarized with fixed threshold) using Image J (*Schneider et al., 2012*)(RRID: SCR_003070). Pyknotic nuclei (defined as nuclei of less than or equal to 450 adjoining pixel units four pixel/μm) were numerated using the Analyse Particle function and expressed as a percentage of the number of total cells.

## Immunohystochemistry

For the in vitro analysis of Oxtr expression using $Oxtr^{Vn/+}$ mice (*Figure 4* and *Figure 4—figure supplements 1* and *2*) 3 weeks-old animals were deeply anesthetized by intraperitoneal administration of 4 mg Avertin [2,2,2-tribromoethanol (99% v/v); dissolved in 2-methylbutan-2-ol) and perfused intracardially using a peristaltic pump for 2 min with ice-cold PBS, followed by 5 min with 4% (wt/vol) paraformaldehyde/PB. The brains were then isolated and post-fixed for 16 hr at 4°C with 4% paraformaldehyde. After extensive washes with PBS, the brains were passed through a sucrose gradient. The brains were then snap frozen in a −45°C isopentane bath, and coronal cryosections of 50 μm thickness were collected on Superfrost Plus slides, and air dried for 1 hr. After washing with PBS, brain sections were incubated with gentle agitation using blocking solution (10% goat serum, 0.3% Triton X-100 in PBS) and subsequently incubated at 4°C overnight with primary antibodies and DAPI diluted in blocking buffer, followed by incubation with the corresponding secondary antibodies conjugated with appropriate fluorophores for 4 hr at RT. Subsequently, the sections were washed three times with PBS and mounted on slides using Immu-Mount (Thermo Scientific) mounting medium. Fluorescence images were acquired using a Leica SP2 confocal microscope using a 10x objective and a 40x objective for higher magnification. For quantitative analyses of the number of Oxtr-Venus neurons co-localizing with GAD67 or GABA (*Figure 4*), the number of Oxtr Venus-positive neurons was estimated on thresholded, binarized images using the Analyze Particle tool in ImageJ, while the number of colocalizing cells was counted manually with the help of the Cell Counter plugin in ImageJ. Imaging of WT and $Oxtr^{Vn/Vn}$ hippocampal neurons expressing tdTomato in coronal sections was performed with a Leica SP5 confocal microscope by acquisition of tdTomato fluorescence in z-stack serial scans at 0.1 μm steps. The morphological analyses (*Figure 6*) were performed manually on z-stack maximum projections. CA1 hippocampal pyramidal neurons, whose polarized structure was maintained in coronal sections, were analyzed by discriminating between the basal neurites (this terminology includes both dendrites and the axonal projection) and the apical dendrites of electroporated cells. On the contrary, since the entire polarized structure of CA3 neurons is not preserved/included upon coronal cryo-sectioning, the whole complexity of primary neurites was analyzed. Twenty micrometer $Oxtr^{-/-}$ and $Oxtr^{+/+}$ brain sections (*Figure 5—figure supplement 1*) from 3-months-old animals were imaged using a Zeiss LSM 710 laser scanning confocal microscope. PV-positive cells were counted separately for the two hemispheres and normalized for a defined area. Analysis was performed using the ImageJ software. For spine and morphological analysis of $Oxtr^{Vn/Vn}$ and $Oxtr^{+/+}$ hippocampal slices (*Figure 8*), CA1 and CA3 biocytin-filled pyramidal neurons were used. Cells were voltage clamped at −70 mV for 20 min in order to allow the diffusion of the pipette solution containing biocytin to the dendtritic compartment. After carefully removing the patch pipette at the end of the recording, hippocampal slices were first fixed in 4% paraformaldehyde and then incubated at room temperature for 4 hr with Alexa-555-labeled streptavidin (Molecular Probes) and DAPI, diluted in PBS containing 10% goat serum and 0.3% Triton X-100. After repetitive washing with PBS, slices were mounted on slides in polyethylene glycol mounting media (Aqua Poly Mount 18606; Polysciences). For dendrite analysis, images were taken using a Leica SP5 confocal microscope (Leica) with a 40x oil objective. Using the NeuronStudio software (NIC, Mount Sinai

School of Medicine, New York, NY, USA; RRID:SCR_013798) (*Rodriguez et al., 2008*, *2006*; *Wearne et al., 2005*), we first performed a semi-automatic tracing of the entire dendritic tree and then used the NeuronStudio segmentation data (*Wearne et al., 2005*) to count the cumulative branching points. For spine analysis fluorescent neurons were imaged using a Leica SP5 microscope with a 8 kHz resonant scanner, a 100 x/1.44 NA oil objective (Leica HCX PL APO CS; Leica), and a white light laser light source (SuperK EXTREME EXW-12, NKT Photonics A/S). Fluorescence signals were excited at 555 nm, and emission was acquired between 565 nm and 650 nm. Single CA3 pyramidal neurons were imaged entirely in a single stack of images (step size 0.13 μm) with the stitching extension provided by the Leica software (Leica APS-AF). Following acquisition, all images were filtered using the 3D Gaussian filter in Image J (σx,y 0.7, σz 1.0) with the help of a custom-written macro to process the large data set. Dendritic spines were counted and classified into stubby, thin, and mushroom-type using the NeuronStudio segmentation algorithm by keeping the suggested parameters (*Rodriguez et al., 2008*), except for the 'threshold correction', which we set to 20%, and the head diameter of mushroom spines, which we set to 0.3 μm.

## Immunocytochemistry

Autaptic cultured neurons were fixed at 14 DIV by incubation in a solution containing 4% paraformaldehyde (w/v)/4% sucrose (w/v) in phosphate-buffered saline, pH 7.4, for 20 min. Subsequently, cells were incubated for 20 min with gentle agitation in blocking solution containing 0.3% (v/v) Triton, 10% (v/v) normal goat serum (GIBCO), and 0.1% (w/v) fish skin gelatin (Sigma) in PBS. The same solution was used for diluting primary and secondary antibodies. Neurons were incubated with primary antibodies for 2 hr at room temperature or overnight at 4°C, followed by incubation with the corresponding secondary antibodies conjugated with appropriate fluorophores for 2 hr at RT. After repetitive washes with PBS, coverslips were mounted using the Vectashield medium with DAPI (Vector Laboratories). The same protocol was followed for all immunolabeling, with the exception of the staining for PSD95, for which cells were fixed in absolute methanol for 7 min at −20°C. Immunofluorescence staining of hippocampal mass cultures (*Figure 5* and *Figure 5—figure supplements 1* and *2*) was carried out as described previously (*Lentini et al., 2008*). Briefly, following fixation, neurons were incubated for 20 min in gelatin dilution buffer [GDB, 0.02 M sodium phosphate buffer, pH 7.4, containing 0.45 M NaCl, 0.2% (w/v) gelatin] containing 0.3% (v/v) Triton X-100, and incubated with primary antibody in GDB for 2 hr at RT. After washing with PBS, coverslips were incubated for 1 hr at RT with secondary antibodies. Nuclei were stained following an incubation for 5 min with 1 μM DAPI in PBS. For the analysis of inhibitory and excitatory synapses in autaptic cultured neurons, triple stainings for Gephyrin, VGAT, and Map2, or for PSD95, VGLUT1, and Map2, were performed. Image acquisition was performed using a Leica SP2 confocal microscope at high magnification (40x objective, numerical aperture 1.0; resolution 1024 × 1024 pixels), and image analysis was carried out using the ImageJ software (NIH). Synapse number was assessed as previously reported (*Varoqueaux et al., 2002*). Briefly, the fluorescent signals of pre- and post-synaptic markers were first thresholded, and then the number of puncta in each cell was measured after the application of a separation filter capable of distinguishing larger puncta. In colocalization experiments assessing immunostained pre- and post-synaptic proteins, the background signal was substracted using the Subtract Background Plugin, and the Intensity Correlation Analysis plugin was used to calculate the Manders' overlap coefficient. Single cell dendrite complexity was evaluated using the advanced Sholl analysis plugin of ImageJ, in which concentric circles are drawn at 5 μm intervals around a common center in the cell body, and the numbers of crossing dendrites are counted at each circle (*Sholl, 1953*). From the Sholl analysis data, we then calculated the number of intersecting dendrites in the interval, expressed as distance from the cell body, showing the largest number of intersections (≤25% below the maximum number of intersections). This interval was 25–80 μm in hippocampal neurons and 15–60 μm in striatal cultures. The images used for the Sholl analysis were acquired using an upright epifluorescence microscope (Zeiss Axio Imager Z1) with a 25x objective lens. For the analysis of pre- and post-synaptic proteins in hippocampal mass cultures (*Figure 5* and *Figure 5—figure supplement 2*), stained neurons were subjected to imaging analysis using a Zeiss LSM 710 laser scanning confocal microscope with a 63x objective. For each coverslip (two for each experimental condition) from 3–4 independent cell preparations, eight random fields were captured and analyzed. For VGAT and VGLUT1 experiments, z-stack serial scans at 1 μm steps were taken and the puncta juxtaposed to Map2-positive processes with different lengths were counted and normalized to 10

μm dendrite length. The quantitative analysis of post-synaptic markers (NL1, NL2, PSD95, Gephyrin, GluA1, GluN1, and GABAγ2) was performed by calculating three distinct parameters, i.e. the total number of merged fluorescent clusters, and the intensities and areas of fluorescence in dendrites with different lengths. Only the fluorescent cluster density is shown here for reasons of brevity because the other two parameters gave identical results. The data were then normalized to 10 μm dendritic length. In each field, 7–8 dendritic tracts with different lengths were analyzed up to a total of 105/120 dendrites for each of the three preparations using the NIH ImageJ software.

## Antibodies

For autaptic culture staining, the following primary antibodies were used: Rabbit polyclonal antibodies directed against VGAT (1:1000, Synaptic Systems, RRID:AB_887871), VGLUT1 (1:1000, Synaptic Systems, RRID:AB_887875), and Map2 (1:500, Chemicon, RRID:AB_91939); chicken polyclonal antibody against Map2 (1:500, Novus, RRID:AB_2138178); mouse monoclonal antibodies against Gephyrin (3B11, 1:500, Synaptic Systems, RRID:AB_887719), and PSD95 (6G6-1C9, 1:200, Abcam, RRID:AB_303248). For immunofluorescence analysis of mass hippocampal cultures, the following primary antibodies were used: Mouse monoclonal antibodies against GAD67 (MAB5406, 1:1000, Chemicon, RRID:AB_2278725); VGAT (1:100, Synaptic Systems, RRID:AB_887872), Gephyrin (hybridoma supernatant, 1:200, Synaptic Systems, RRID:AB_2232546), VGLUT1 (1:150, Synaptic Systems, 135 311 RRID:AB_887880), GluN1 (1:150, Synaptic Systems, RRID:AB_887750), Synaptophysin (1:200, Synaptic Systems, RRID:AB_887822), and PSD-95 (1: 150, Thermo Scientific, AB_325399); rabbit polyclonal antibodies against GFP (1:1000, MBL, RRID:AB_591819), VGLUT1 (1:150, Synaptic Systems, RRID:AB_887875), NL2 (1:50, Synaptic Systems, RRID:AB_993014), GABA γ2A (1:500, Synaptic Systems, RRID:AB_2263066), Synaptophysin (1:100, Synaptic Systems, RRID:AB_887905), Parvalbumin (1:100, Synaptic Systems, RRID:AB_2156474), GluA1 (1:150, Abcam, RRID:AB_2113447), Map2 (1:500, Chemicon, RRID:AB_91939), GAD65/67 (1:200, Chemicon, RRID:AB_11210186), and NL1 (1:50, Santa Cruz Biotechnology, AB_2151659). For the immunohistochemical analysis of brain sections, the following primary antibodies were used: Rabbit polyclonal antibody against GFP (1:1000, MBL, RRID:AB_591819), Parvalbumin (1:200, Synaptic Systems, RRID:AB_2156474), S100 (1:100; DAKO, RRID:AB_10013383), GABA (1:500, A2052 Sigma, RRID:AB_477652); mouse monoclonal antibodies against GAD67 (1:1000, MAB5406, Chemicon, RRID:AB_2278725), and NeuN (1:2000, Chemicon, RRID:AB_2298772); chicken polyclonal antibody against Map2 (1:500, Novus Biologicals, RRID:AB_2138178) and GFP (1020, 1:500; Aves, RRID:AB_10000240).

## *In utero* electroporation

Hippocampal progenitors at embryonic stage 14.5 (E14.5) were transfected *in utero* with a plasmid encoding a membrane-targeted mutant of tdTomato. For morphological analyses, littermates from $Oxtr^{Vn/+}$ interbreedings were used. *In utero* electroporation to the ammonic neuroepithelium resulted in the specific and sparse targeting of neuronal progenitors of pyramidal neurons which allowed the reconstruction and subsequent morphological analysis of isolated nerve cells. Pregnant females were deeply anesthetized and kept on a warming pad (31–32°C) during the entire surgical procedure. The anesthesia was maintained by a steady supply of isoflurane (DeltaSelect) and oxygen (1 L/min). The uterus with the embryos was exposed through a ~2 cm midline incision in the ventral abdomen peritoneum and moistened with warmed PBS containing antibiotics (1000 U/ml penicillin/streptomycin; Invitrogen). The plasmid [at a concentration of 1.25 μg/μl, prepared using an Endo-Free plasmid purification kit (Quiagen) and dissolved in TE (TrisHCl pH 8.0) buffer with 0.05% Fast Green dye (Sigma)] was injected using a glass capillary into the lateral ventricles of embryos. Electric square pulses (38 V, 50 mA, 50 ms duration, 950 ms intervals, six pulses per embryo) were applied to each embryo with an ECM830 electroporator (BTX Harvard Apparatus). Subsequently, the uterus was placed back into the abdominal cavity, and the body wall was sutured. P7 animals were used for neuronal morphology reconstruction. After intracardiac perfusion with saline followed by 4% PFA, brains were isolated from skulls and post-fixed for 16 hr in 4% PFA in phosphate buffer (PB; pH 7.4), followed by extensive washing with PBS. After passing through a 10–30% sucrose gradient, 100 μm-thick coronal cryosections were collected on Superfrost Plus glass slides (Thermo Scientific), mounted, and signals of tdTomato fluorescence were acquired with a Leica SP5 confocal microscope using a 40x objective.

## Electrophysiology

For electrophysiology, autaptic cultured neurons (9–14 DIV) were whole-cell voltage-clamped at −70 mV with an EPSC10 amplifier (HEKA) under the control of the Patchmaster two program (HEKA). All traces were analyzed using AxoGraph X (AxoGraph Scientific). The experiments were performed using a patch-pipette solution containing (in mM) 136 KCl, 17.8 Hepes, 1 EGTA, 0.6 $MgCl_2$, 4 NaATP, 0.3 mM $Na_2GTP$, 15 creatine phosphate, and 5 U/mL phosphocreatine kinase (315–320 mOsmol/L, pH 7.4). The extracellular solution used for all recordings contained (in mM) 140 NaCl, 2.4 KCl, 10 Hepes, 10 glucose, 4 $CaCl_2$ and 4 $MgCl_2$ (320 mOsml/liter), pH, 7.3. Evoked EPSCs and IPSCs were induced by depolarizing the cell from −70 to 0 mV at a frequency of 0.2 Hz. The size of the ready releasable pool (RRP) was measured upon 6 s application of 0.5 M hypertonic sucrose solution. The vesicular release probability ($P_{vr}$) was calculated by dividing the charge transfer during an action potential induced PSC by the charge transfer during the sucrose response. Short-term plasticity was evaluated by recording PSCs during 10 Hz stimulation trains. Miniature EPS (mEPSC) and IPS (mIPSC) currents were recorded in the presence of 300 nM tetrodotoxin (TTX, Tocris Bioscience). The cell surface expression of GABA and glutamate receptor was assessed by focal application of 100 µM glutamic acid (Sigma) or 3 µM GABA (Sigma), respectively. Spontaneous postsynaptic currents (sPSCs; *Figure 5* and *Figure 5—figure supplement 2*) were recorded in a whole-cell configuration using an internal solution containing (in mM) 135 CsCl, 3 NaCl, 10 EGTA, 0.5 $CaCl_2$, 1 $MgCl_2$, 4 ATP, 0.3 GTP, and 10 Hepes, pH 7.4, with CsOH, and a bath solution containing (in mM) 140 NaCl, 3 KCl, 10 glucose, 2 $CaCl_2$ and 1 $MgCl_2$, 10 Hepes, pH 7.4, with NaOH. During sPSC recordings, the internal solution was supplemented with 5 mM N-ethyllidocaine (QX-314) in order to block action potentials in the patch-clamped neuron. Moreover, during the recording of glutamatergic PSCs (sEPSC) 10 µM bicuculline was added to block $GABA_A$ receptors, whereas during GABAergic PSC (sIPSC) recording 10 µM 2,3-dihydroxy-6-nitro7-sulfamoyl-benzo[f]quinoxaline-2,3-dione (NBQX) and 30 µM 4- (3phosphonopropyl)piperazine-2-carboxylic acid (CPP) were added to block AMPA and NMDA receptors, respectively. Spontaneous synaptic events were detected using NeuroMatic, a collection of Igor Pro (WaveMetrics Inc.) functions (http://www.neuromatic.thinkrandom.com/) upon setting at 10 pA as detection threshold. The amplitude averages and standard deviations were calculated using the lognormal function of the Origin 7.5 software (OriginLab Corp), due to lognormal distribution of the amplitudes of spontaneous synaptic events. Inter-event intervals for spontaneous synaptic events were distributed exponentially and mean intervals were obtained from the best mono-exponential fit of the inter-event interval distributions. Gamma oscillations were recorded in juvenile $Oxtr^{+/+}$ and $Oxtr^{Vn/Vn}$ mice (P15–26). Transverse hippocampal slices (300 µm thickness) were prepared as described previously (*Bischofberger et al., 2006*) using a vibratome (Leica VT1200S) from age-matched littermates anesthetized with isoflurane (DeltaSelect). Brain isolation and slice preparation were performed on ice in a sucrose-based slicing solution (230 mM sucrose, 26 mM $NaHCO_3$, 2 mM KCl, 1 mM $KH_2PO_4$, 2 mM $MgCl_2$, 10 mM D-glucose, 0.5 mM $CaCl_2$) under oxygenation with carbogen gas (95% $O_2$, 5% $CO_2$). After sectioning, slices were placed in a holding chamber containing ACSF (120 mM NaCl, 26 mM $NaHCO_3$, 1 mM $KH_2PO_4$, 2 mM KCl, 1 mM $MgCl_2$, 10 mM D-glucose, 2 mM $CaCl_2$) and allowed to recover for 10–15 min before recording. Hippocampal γ-oscillations were recorded in the CA3 region, where the peak power of kainate-induced γ-oscillations is reported to be the highest (*Craig and McBain, 2015*; *Hammer et al., 2015*). All recordings were performed using an interface recording chamber (BSCBU Base Unit with the BSC-HT Haas Top, Harvard Apparatus). Slices were constantly perfused with ACSF and the temperature was maintained at 33°C. Extracellular recording electrodes filled with ACSF were placed in the pyramidal cell layer of CA3 and extracellular field potentials were recorded for ≥20 min before the addition of 100 nM kainate. Baseline (≥20 min) and oscillatory (30 min) activities were acquired using a 700B amplifier (Axon Instruments, Molecular Devices), low pass Bessel filtered at 3 KHz, and digitized by the Digidata 1440A data acquisition system (Axon Instruments, Molecular Devices). All electrophysiological traces were analyzed using Axograph X software (AxoGraph Scientific). Power spectra were calculated upon Fourier transforms of 10 min epochs (last 10 min of each recording) of recorded field activity. The baseline power spectrum was subtracted from the power spectrum obtained during kainate application. The dominant frequency within 25–45 Hz was used to quantify the peak frequency. The power was calculated as the area under the respective peak in the power spectrum. Whole-cell patch-clamp recordings were performed in $Oxtr^{+/+}$ and $Oxtr^{Vn/Vn}$ hippocampal slices at room

temperature (22°C). Slices were prepared and maintained in ACSF under oxygenation with carbogen gas (95% $O_2$, 5% $CO_2$), as described in the previous paragraph. The holding potential was set at −70 mV using an EPSC10 amplifier (HEKA) under the control of the Patchmaster software (HEKA). Miniature post-synaptic currents (mPSCs) were recorded in the presence of 1 μM TTX from CA1 and CA3 neurons showing the typical morphological properties of pyramidal cells. In addition to TTX, miniature IPSCs (mIPSCs) were recorded in the presence of 10 μM 6-cyano-7- nitroquinoxaline-2,3-dione (CNQX), and 20 μM D-2-amino-5-phosphonopentanoic acid (D-AP5) to block excitatory post-synaptic currents. Miniature EPSCs (mEPSCs) were recorded in the presence of 10 μM bicuculline methiodide to block inhibitory postsynaptic currents. Two different intracellular solutions were used during mIPSCs and mEPSCs acquisitions. The pipette solution for recording mIPSCs contained 0.4% biocytine, 120 mM potassium gluconate, 20 mM KCl, 10 mM EGTA, 2 mM $MgCl_2$, 2 mM $Na_2ATP$, and 10 HEPES, pH adjusted to 7.28 with KOH, osmolarity adjusted to 315 mOsm/L with sucrose. The pipette solution for recording mEPSCs contained contained 0.4% biocytine, 140 mM KCl, 10 mM EGTA, 2 mM $MgCl_2$, 2 mM $Na_2ATP$, and 10 HEPES, pH adjusted to 7.2 with KOH, osmolarity adjusted to 315 mOsm/L with sucrose. All electrophysiological traces were analyzed using Axograph X software (AxoGraph Scientific) and miniature events were captured using the template matching algorithm provided by the software.

## Data analysis and statistics

All results are shown as mean ± SEM. Statistical significance was assessed using one-way ANOVA followed by post-hoc Bonferroni test or by two-tailed Student's t-test, where appropriate. Statistical analyses were performed using the Prism version five software (GraphPad, San Diego, CA, USA).

## Acknowledgements

This work was supported by the Cariplo Foundation (Grant 2008.2314, MP, MT), the Max Planck Society (NB, JSR), the European Commission (COSYN, JSR, NB), the Fritz Thyssen Foundation (HK), the German Research Foundation (CNMPB, NB; SPP1365/KA3423/1–1, HK and NB; KA3423/3–1, HK), and an 'Integrated research on neuropsychiatric disorder' grant in the Strategic Research Program for Brain Sciences by the Ministry of Education, Culture, Sports, Science, and Technology of Japan (KN). We thank V Tarabykin and C Rosemund (Charité, Berlin, Germany) for expression plasmids, H Taschenberger for advice regarding electrophysiological recordings in acute slices, and A Günter, S Bolte, I Beulshausen, M Schwark, and the staff of the animal facility and DNA core facility at the Max Planck Institute of Experimental Medicine for excellent technical support.

## Additional information

### Funding

| Funder | Grant reference number | Author |
|---|---|---|
| Fondazione Cariplo | Grant 2008.2314 | Marco Parenti |
| Max-Planck-Gesellschaft | | JeongSeop Rhee |
| European Commission | EU-AIMS FP7-115300 | Nils Brose |
| Fritz Thyssen Stiftung | | Hiroshi Kawabe |
| Deutsche Forschungsgemeinschaft | CNMPB | Nils Brose |
| Deutsche Forschungsgemeinschaft | SPP1365/KA3423/1-1 | Nils Brose |
| Deutsche Forschungsgemeinschaft | KA3423/3-1 | Hiroshi Kawabe |
| Ministry of Education, Culture, Sports, Science, and Technology | | Katsuhiko Nishimori |

The funders had no role in study design, data collection and interpretation, or the decision to submit the work for publication.

## Author contributions

SR, Conceptualization, Data curation, Methodology, Writing—original draft, Writing—review and editing; MCA, Data curation, Formal analysis, Methodology; FG, MG, GB, MH, MT, Data curation, Formal analysis; IB, Resources, he provide the plasmids encoding tdTomato; LPT, Data curation, Formal analysis, Methodology, Writing—review and editing; AS, He provided a macro for analysis and took images with confocal microscope for spine analysis; HK, Resources, Validation; KN, Resources, he provided the two mice lines for this project, Oxtr -/- and Oxtr vn/vn; NB, Funding acquisition, Writing—original draft, Writing—review and editing; MP, Conceptualization, Funding acquisition, Writing—original draft; JSR, Conceptualization, Supervision, Funding acquisition, Writing—original draft, Project administration, Writing—review and editing

## Author ORCIDs

JeongSeop Rhee, http://orcid.org/0000-0002-8560-3630

## Ethics

Animal experimentation: All animal experiments were performed in accordance with the guidelines for the welfare of experimental animals issued by the State Government of Lower Saxony, Germany, and the Italian Government in compliance with European and NIH guidelines. (33.9-42502-04-13/1359; 33.9-42502-04-13/1052).

## Additional files

### Supplementary files

• Supplementary file 1. Summary of the synaptic transmission analysis. The Table provides an overview of all the electrophysiological experiments performed in this paper. The conditions used for each experiment are reported together with the related figures. Data are expressed as mean ± SEM. Fs: figure supplement.

• Supplementary file 2. Summary of the morphological analysis. The Table provides an overview of all the morphological experiments performed in this paper. The conditions used for each experiment are reported together with the related figures. Data are expressed as mean ± SEM. Fs: figure supplement.

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
