## [Decision Letter]

Thank you for submitting your article "Transient Oxytocin Signaling Primes the Development and Function of Excitatory Hippocampal Neurons" for consideration by *eLife*. Your article has been favorably evaluated by a Senior Editor and three reviewers, one of whom is a member of our Board of Reviewing Editors. The reviewers have opted to remain anonymous.

The reviewers have discussed the reviews with one another and the Reviewing Editor has drafted this decision to help you prepare a revised submission. We hope you will be able to submit the revised version within two months.

Summary:

This manuscript by Ripamonti et al. reports the role of oxytocin in the regulation of dendritic complexity, synapse density, and synaptic transmission. In support of this conclusion, the authors demonstrate that oxytocin treatment of hippocampal autaptic culture during DIV1-3 induces at a later stage (DIV14) decreases in the dendritic complexity and excitatory synaptic transmission in glutamatergic neurons but no changes in inhibitory synaptic transmission in GABAergic neurons. This difference appears to be attributable to low levels of oxytocin receptors in GABAergic neurons as supported by the result that exogenous expression of oxytocin receptors in striatal GABAergic neurons renders oxytocin sensitivity and related changes in synapses/transmission in these neurons. Conversely, in oxytocin receptor-lacking neurons from two different mouse lines, there are increases in dendritic complexity, E-I synapse ratio, and excitatory synaptic transmission.

The suggested roles of oxytocin in the regulation of excitatory neuronal morphology and excitatory synapse development and function are different from the focus of previous works on the role of oxytocin in the regulation of GABAergic neuronal functions. The experimental data are solid and seem to support the main conclusions.

Essential revisions:

1) The authors indicated that the oxytocin surge occurs during the last days of gestation and peaks just before the delivery (subsection “Oxt reduces dendrite branching and function of glutamatergic neurons via G_q/11_-coupled Oxtrs”, first paragraph). If this is true, why did the authors prepare hippocampal cultures using P0 pups, which may already be exposed to oxytocin surge?

2) Is 100nM Oxt physiologically relevant? Also, is Oxtr expressed at 1-3DIV neurons? How about astrocytes? Do they express Oxtr as well?

3) The neuronal cultures from *Oxtr^-/-^* mice were prepared from E18, while other cultures were prepared using P0 pups. Why is there discrepancy? Also, it will be nice to perform oxytocin treatment on DIV1-3 *Oxyt^-/-^* neurons as a control.

4) Exogenous oxytocin treatment resulted in severe dendritic atrophy in pyramidal neurons. The authors may want to test neuronal viability or apoptosis upon oxytocin treatment.

---

## [Author Response]

*Essential revisions:*

*1) The authors indicated that the oxytocin surge occurs during the last days of gestation and peaks just before the delivery (subsection “Oxt reduces dendrite branching and function of glutamatergic neurons via G_q/11_-coupled Oxtrs”, first paragraph). If this is true, why did the authors prepare hippocampal cultures using P0 pups, which may already be exposed to oxytocin surge?*

We acknowledge the validity of this comment, which is mainly relevant in the context of explaining the phenotype of Oxtr-deficient neurons that were cultured at E18 (Figure 5). In our Figure 5—figure supplement 2 we demonstrate that the phenomena seen in autaptic cultures prepared from P0 pups and treated with Oxt are corroborated by data on neurons in mass cultures prepared from E18 embryos and exposed to 10 nM TGOT, a selective Oxtr agonist.

Despite these arguments, we acted upon this comment and studied autaptic hippocampal cultures from E18 mouse embryos to complement the data obtained with P0 autaptic cultures. The results obtained on E18 cultures are very similar to those obtained on P0 autaptic cultures: Oxt exposure reduced the dendrite complexity of hippocampal neurons and impaired glutamatergic synaptic transmission. Our new data indicate that neurons cultured at P0 remain sensitive to Oxt after culturing, even though they must have been exposed to Oxt shortly before and during parturition, indicating that Oxt signalling towards these neurons has not reached saturation – or that the culturing process causes some type of 'reset' in the neurons. The new data on the effects of Oxt on neuronal morphology and synaptic transmission in E18 cultures are now shown in Figure 2—figure supplement 3, and described/discussed in the main text (subsection “Oxt reduces dendrite branching and function of glutamatergic neurons via G_q/11_-coupled Oxtrs”, sixth paragraph). Given that the reviewers requested us to shorten the manuscript somewhat, we abstained from a detailed further discussion of this issue.

*2) Is 100nM Oxt physiologically relevant? Also, is Oxtr expressed at 1-3DIV neurons? How about astrocytes? Do they express Oxtr as well?*

In essence, there is currently no way to tell the exact Oxt concentration during physiologically relevant Oxt signalling in the hippocampus, which receives Oxt via direct projections and diffusion from other more distant release sites. Reported concentrations of Oxt in extracellular compartments in the brain, CSF, or plasma vary substantially depending on the study at hand. However, it is likely that resting physiological concentrations of Oxt are below 100 nM. In normal, non-pregnant mice and rats, Oxt levels in microdialysates (30 min dialysis at 0.33 microliter/min) from amygdala and hippocampus are in the range of 50-100 pM. However, these values are likely substantially lower than the actual extracellular Oxt concentration. During pregnancy, corresponding Oxt levels in plasma (and hence in the embryos) can increase manifold. As explained in our paper, we chose an Oxt concentration that is actually substantially lower than the concentrations used in other very prominently published papers on Oxt effect on neurons – 100 nM. This concentration was chosen based on the criteria of (i) maximal effect on Oxt receptors (Kd of about 1 nM for Oxt) and (ii) minimal cross-reactivity with V1a receptors (Kd about 20-30 nM for Oxt) (Busnelli et al., 2013, J Pharmacol Exp Ther 346, 318-327). Experts in the specificity titration of Oxt concentrations use the same concentrations for the treatment of hippocampal neurons as we do (Leonzino et al., 2016, Cell Rep 15, 96-103). This issue is now discussed in the revised version of the paper (subsection “Mass culture of hippocampal neurons and autaptic neuronal culture”).

Hippocampal neurons in culture express Oxt receptors already at DIV1, and expression increases over time in culture (Leonzino et al., 2016, Cell Rep 15, 96-103). Hence, at the time of our treatment, Oxt receptors are already expressed in hippocampal neurons. This information is now provided in the revised manuscript (subsection “Oxt reduces dendrite branching and function of glutamatergic neurons via _Gq/11_-coupled Oxtrs”, second paragraph).

Another important new issue in this context is that Oxt receptor knock out neurons do not response to our Oxt treatment – see also our comment to point 3 below – indicating that the effects we describe are specifically caused by Oxt receptor activation. These new data are now shown in Figure 1—figure supplement 3, and described/discussed in the main text (subsection “Oxt reduces dendrite branching and function of glutamatergic neurons via G_q/11_-coupled Oxtrs”, first and fifth paragraphs).

As regards the involvement of astrocytes in the Oxt effects we see, we do not think that they play a significant role because we show that selective expression of Oxt receptors in inhibitory neurons (under a synapsin promoter) makes these cells sensitive to Oxt. Further, our labelling of cells expressing Oxt receptors (Figure 4 and Figure 4—figure supplement 1) does not detect cells with astrocyte-like appearance and localization but a nice colocalization with NeuN, indicating that it is mainly neurons that express Oxt receptors.

Despite these arguments, we acted upon the corresponding comment of the reviewers. We directly assessed Oxt receptor expression in astrocytes in *Oxtr^Vn/+^*hippocampus and cortex. No colocalization was observed between Venus, which reports Oxtr expression and the astrocyte marker S100. These were complemented by Western blotting experiments with astrocyte and whole brain extracts using antibodies to Venus and the astrocyte marker GFAP. Here, brain samples were shown to contain both GFAP and Venus, while no Oxtr expression was detected in primary cultured *Oxtr^Vn/Vn^* or *Oxtr^Vn/+^*astrocytes. These new data are now shown in Figure 4—figure supplement 2, and described/discussed in the main text (subsection “Inhibitory hippocampal neurons are less affected by Oxt because of low Oxtr expression levels”, third paragraph).

*3) The neuronal cultures from Oxtr^-/-^ mice were prepared from E18, while other cultures were prepared using P0 pups. Why is there discrepancy? Also, it will be nice to perform oxytocin treatment on DIV1-3 Oxyt^-/-^ neurons as a control.*

Taking embryos at P0 is technically easier and saves the mothers for further breeding. Hence, experiments on WT cells were done using tissue from P0 pups. On the other hand, offspring from Oxt receptor KO dams, which were used in the experiments shown in Figure 5, die usually quickly after birth due to aberrant maternal care (Takayanagi et al., 2005, Proc Natl Acad Sci USA 102, 16096-16101). To prevent deteriorating effects on the pups in these experiments, we decided to routinely take the embryos at E18. This information is now provided in the main text (subsection “Mass culture of hippocampal neurons and autaptic neuronal culture”).Please see also our response to point 1 above.

As suggested by the reviewers, we performed new control experiments by treating Oxt receptor knock out neurons with Oxt for 1 or 3 days. No differences were found between control and Oxt treated groups indicating that the Oxt effects we observed in hippocampal neurons are selectively mediated by Oxt receptor. These new data are now shown in Figure 1—figure supplement 3, and described/discussed in the main text (subsection “Oxt reduces dendrite branching and function of glutamatergic neurons via G_q/11_-coupled Oxtrs”, first and fifth paragraphs).

*4) Exogenous oxytocin treatment resulted in severe dendritic atrophy in pyramidal neurons. The authors may want to test neuronal viability or apoptosis upon oxytocin treatment.*

We concur with the notion that it would be interesting to test neuronal viability or apoptosis upon Oxt treatment. Hence, we analysed neuronal survival in hippocampal mass cultures upon Oxt treatment by firstly analysing nuclear pyknosis (nuclear condensation) and, additionally, by determining the percentage of cells that are propidium iodide positive (DNA fragmentation) in control and Oxt-exposed cultures (Tuffy et al., 2010, Mol Cell Biol 30, 5484-5501). No significant differences were found between control and treated groups in the percentage of pyknotic nuclei, indicating that our Oxt treatment does not affect cell viability. These new data are now shown in Figure 1—figure supplement 1, and described/discussed in the main text (subsection “Oxt reduces dendrite branching and function of glutamatergic neurons via G_q/11_-coupled Oxtrs”, first paragraph).

In the first set of experiments we also attempted a co-staining with propidium iodide and DAPI. Hippocampal neurons were first incubated with propidium iodide (PI, 5 μM) in culture medium for 30 min, and upon fixation, nuclei were stained with DAPI. However, due to the fact that PI stained not only the nuclei but also the cell bodies, we determined that the PI had leaked outside the nucleus, most likely due to the fixation method employed (see Figure 9). Therefore, we performed a new set of experiments where neurons were incubated with Hoechst 33342 (1 μg/ml) and propidium iodide (5 μM) in culture medium for 30 min, directly mounted on glass slides, and imaged for image analysis. Even though in E18 Oxt treated cultures the percentage of PI-positive cells appeared to be slightly reduced as compared to the control group (Figure 9), we abstained from following this up further due to the time limitation for resubmission (currently, the data represent two independent cultures prepared from E18 embryos and only one P0 neuronal preparation). In any case, even the data in Figure 9 indicate that Oxt treatment does not cause cell death.

Author response image 1.Assessment of apoptosis using propidium iodide.(**A**) Representative images of propidium iodide (PI)-positive nuclei and Hoechst-labeled nuclei captured immediately after mounting after fixation under different conditions. (**B**) Representative images of propidium iodide (PI)-positive nuclei and Hoechst-labeled nuclei captured 48 hr after mounting after fixation under different conditions. (**C**) PI-positive neurons were quantified in separate channels and expressed as a percentage of total cells assessed by Hoechst staining in the same field. Data are expressed as mean ± SD.**DOI:**
http://dx.doi.org/10.7554/eLife.22466.025